# Ferric reductase-related proteins mediate fungal heme acquisition

Udita Roy[1], Shir Yaish[1], Ziva Weissman[1], Mariel Pinsky[1], Sunanda Dey[1], Guy Horev[2], Daniel Kornitzer[1]*

[1]Department of Molecular Microbiology, B. Rappaport Faculty of Medicine, Technion - Israel Institute of Technology, Haifa, Israel; [2]Bioinformatics Knowledge Unit, Technion - Israel Institute of Technology, Haifa, Israel

**Abstract** Heme can serve as iron source in many environments, including the iron-poor animal host environment. The fungal pathobiont *Candida albicans* expresses a family of extracellular CFEM hemophores that capture heme from host proteins and transfer it across the cell wall to the cell membrane, to be endocytosed and utilized as heme or iron source. Here, we identified Frp1 and Frp2, two ferric reductase (FRE)-related proteins that lack an extracellular N-terminal substrate-binding domain, as being required for hemoglobin heme utilization and for sensitivity to toxic heme analogs. Frp1 and Frp2 redistribute to the plasma membrane in the presence of hemin, consistent with a direct role in heme trafficking. Expression of Frp1 with the CFEM hemophore Pga7 can promote heme utilization in *Saccharomyces cerevisiae* as well, confirming the functional interaction between these proteins. Sequence and structure comparison reveals that the CFEM hemophores are related to the FRE substrate-binding domain that is missing in Frp1/2. We conclude that Frp1/2 and the CFEM hemophores form a functional complex that evolved from FREs to enable extracellular heme uptake.

**\*For correspondence:**
danielk@technion.ac.il

**Competing interest:** The authors declare that no competing interests exist.

## Editor's evaluation

This work focuses on the important problem of how a human pathogenic fungus obtains iron during infection. This study uses biochemical and genetic methods to identify the missing link between heme receptors and heme utilization by cells. Specifically, they provide convincing evidence that the ferric reductase-like proteins Frp1 and Frp2 have major roles in iron acquisition from heme, thus identifying a new function for these proteins. Understanding how pathogenic fungi obtain iron and identifying differences in fungal and host iron metabolism can provide valuable leads for drug discovery.

## Introduction

*Candida albicans* is a commensal microorganism of the human host (*Ghannoum et al., 2010*; *Nash et al., 2017*). It is normally found on mucosal surfaces and on the skin as part of the normal microbial flora. It is also one of the most common human fungal pathogens, causing infections that range from superficial infections such as vulvovaginal candidiasis (*Sobel, 2007*) to life-threatening systemic infections when the immune system is weakened or when there is damage to epithelial barriers (*Kullberg and Arendrup, 2015*). *Candida* species, of which the most prevalent is *C. albicans*, are responsible for an estimated 750,000 cases of systemic candidiasis annually, with a mortality rate of around 40% (*Bongomin et al., 2017*; *Pfaller et al., 2012*).

Like most organisms, *C. albicans* requires iron for its survival. Iron can be found in several oxidation states, of which the more common are $Fe^{2+}$ and $Fe^{3+}$. The low solubility of ferric iron ($Fe^{3+}$) makes iron

**eLife digest** Hosts and disease-causing fungi are often locked into a battle over resources. The host will attempt to withhold molecules that the fungus needs to survive, while the pathogen will try to find alternative routes to obtain them. *Candida albicans*, for example, can go after the atoms of iron embedded in the proteins of the organism it infects. To do so it releases molecules known as hemophores, which scavenge the iron-containing heme molecule that equips oxygen-carrying proteins in the blood.

Once captured, the heme is carried across the wall that protects *C. albicans* from the environment and brought to the membrane of the cell. It is then taken in and trafficked inside vesicles to its destination. However, the identity of the molecular actors which help to bridge the internal and external segments of the heme journey remain unclear. Previous studies have shown that the hemophore Pga7 is involved, but this protein is attached to the outside of the cell membrane, where it cannot directly interact with the import machinery.

Roy et al. set out to discover this missing link. Examining the genomes of fungal species related to *C. albicans* highlighted two membrane proteins, Frp1 and Frp2, which could participate in heme uptake. Protein sequence comparison revealed that Frp1 and Frp2 were closely related to ferric reductases, a group of membrane enzymes which can chemically alter extracellular iron prior to uptake.

Deleting the genes for Frp1 and Frp2 rendered *C. albicans* cells incapable of taking in heme. Conversely, a fungal species which cannot normally uptake heme could efficiently internalise these complexes when artificially equipped with Frp1 and Pga7, suggesting that the two proteins work closely together. Finally, protein structure comparisons highlighted that an extracellular domain present in ferric reductases but absent in Frp1 and Frp2 is, in fact, related to Pga7 and other hemophores. This implies that the iron and heme uptake systems may share a common evolutionary origin. Overall, the work by Roy et al. reveals a new family of proteins which allow disease-causing fungi to steal iron from their hosts. This knowledge may be useful to design better anti-fungal treatments.

acquisition challenging in oxidizing environments. Many unicellular organisms express ferric reductases (FREs) on their surface, which can contribute to iron acquisition by reducing ferric iron to its more soluble ferrous ($Fe^{2+}$) ion and/or extracting it from iron chelates (*Kornitzer, 2009*; *Kosman, 2003*; *Philpott, 2006*). The *C. albicans* genome contains at least 15 genes encoding FRE-like proteins, including Fre1/Cfl1 and the major iron reductase Fre10 (*Baek et al., 2008*; *Hammacott et al., 2000*; *Knight et al., 2005*).

Heme, or Fe-protoporphyrin IX, is a molecule that serves as a cellular cofactor almost universally across life. As such, it is present in many natural environments, and thus, environmental heme can be utilized as an alternative iron source by many organisms, via a large repertoire of heme acquisition systems (*Donegan et al., 2019*). In animals, HRG-1-like heme transporters conserved from *C. elegans* to mammals can mediate extracellular heme import or intracellular heme transport (*Rajagopal et al., 2008*; *Reddi and Hamza, 2016*) and the human membrane protein FLVCR2 functions as a heme importer (*Duffy et al., 2010*).

For microorganisms, the animal host environment poses a particular challenge for iron acquisition, as the host restricts the availability of free iron by a variety of active mechanisms as part of a defense strategy called 'nutritional immunity' (*Ganz, 2018*; *Weinberg, 1975*). In this environment, hemoglobin heme, comprising about two-thirds of the total iron, is therefore expected to be a particularly attractive iron source, and indeed it is utilized by many bacterial pathogens via several well-studied mechanisms that involve distinct extracellular hemophore families and plasma membrane ABC transporters (*Cassat and Skaar, 2013*; *Contreras et al., 2014*; *Ganz and Nemeth, 2015*; *Hood and Skaar, 2012*; *Huang and Wilks, 2017*).

Fungi have also evolved different mechanisms to internalize heme, involving a membrane heme receptor in the case of *Schizosaccharomyces pombe* (*Labbé et al., 2020*; *Mourer et al., 2015*) and an endocytosis-based mechanism in the case of *Cryptococcus neoformans* (*Bairwa et al., 2019*; *Bairwa et al., 2017*; *Cadieux et al., 2013*). *C. albicans* and other *Candida* spp. express a distinct family of GPI-anchored and soluble extracellular hemophores containing a heme-binding CFEM domain (*Roy and Kornitzer, 2019*). Pga7 and Rbt5 are GPI-anchored to the cell membrane and the cell

wall, respectively, while Csa2 is secreted to the medium (*Kuznets et al., 2014*; *Nasser et al., 2016*; *Weissman and Kornitzer, 2004*). These three CFEM proteins are all able to extract heme from hemoglobin and to transfer it among themselves, consistent with a model where heme captured from hemoglobin or other host heme-binding proteins, such as serum albumin, is transferred across the cell wall to the plasma membrane (*Kuznets et al., 2014*; *Nasser et al., 2016*; *Pinsky et al., 2020*). Heme binding to the CFEM domain involves a unique aspartic acid-mediated heme-iron coordination that renders the heme-binding redox-sensitive: only ferriheme ($Fe^{3+}$) binds to the CFEM domain (*Nasser et al., 2016*).

Heme utilization in *C. albicans* requires, in addition to the CFEM hemophores, components of the endocytic pathway and the ESCRT system, and vacuolar function (*Weissman et al., 2008*), suggesting that it involves endocytosis of the heme to the vacuole. However, GPI-anchored proteins are not known to be subject to ESCRT-mediated endocytosis, suggesting the existence of (an) additional transmembrane protein(s) involved in heme internalization. Here, we describe the identification and characterization of Frp1 and Frp2, plasma membrane proteins related to FREs, that are essential for heme uptake and utilization in *C. albicans*.

## Results

### Phylogenetic profiling for identification of new heme-iron acquisition genes

Analysis of genomes from species related to *C. albicans* indicated that many carry CFEM proteins related to Pga7, Rbt5, and Csa2 (*Nasser et al., 2016*). We found that across fungal genomes of the Ascomycete clades most closely related to *C. albicans*, distinct relatives of Rbt5, Pga7, and Csa2 can be identified in almost every genome (*Figure 1—figure supplement 1* shows a proximity tree of Rbt5-related proteins from 14 Saccharomycetales genomes). One exception is *Meyerozyma* (*Candida*) *guilliermondii*, which completely lacks any coding sequence related to these CFEM proteins, unlike equally distant relatives of *C. albicans* such as *Debaryomyces hansenii* (*Kurtzman and Suzuki, 2010*). In order to identify additional proteins involved in heme uptake, we applied phylogenetic profiling or pathway co-evolution analysis. Briefly, this method is based on the assumption that, as organisms can acquire new functions and evolve new pathways by gene duplication, acquisition and modification, organisms can similarly lose functions, and their associated genes, when they are no longer required. Genes encoding a single functional pathway are thus expected to be preserved or eliminated in a correlated fashion across genomes, according to whether the function is present or absent in a given organism (*Li et al., 2014*; *Pellegrini et al., 1999*). Sequenced fungal genomes currently number well over one thousand, including many tens of Saccharomycetales (*Butler et al., 2009*; *Grigoriev et al., 2014*). Two genes whose presence was best correlated with the presence of Rbt5-related CFEM genes were *FRP1* and *FRP2* (see Materials and methods for details).

*C. albicans* Frp1 and Frp2 were previously classified as FREs (*Almeida et al., 2009*), based on their homology to the main *C. albicans* FRE Fre10 (*Knight and Dancis, 2006*), as well as to Cfl1, another *C. albicans* protein that was experimentally shown to function as an FRE (*Hammacott et al., 2000*). However, alignment followed by phylogenetic tree building indicates that Frp1 and Frp2 cluster separately from most FRE homologs, including Fre10 and Cfl1 (*Figure 1—figure supplement 2A*). Notably, Frp1 and Frp2 lack an N-terminal domain present in most other FREs (*Figure 1—figure supplement 2B*).

Frp1 and Frp2 are 36% identical, and out of 13 Saccharomycetales listed in the Candida Gene Order Browser database (cgob3.ucd.ie; *Fitzpatrick et al., 2010*; *Maguire et al., 2013*) that have Frp homologs, 12 have an Frp1 ortholog, and 10 have an Frp2 ortholog. Interestingly, *FRP1* and *FRP2* are adjacent to two CFEM protein genes in the *C. albicans* genome, *FRP1* to *PGA7* and *FRP2* to *CSA1*. In both cases, the adjacent genes are arranged head-to-head and share a promoter region. This synteny is conserved across most genomes (75% for *FRP1-PGA7* and 70% for *FRP2-CSA1*; *Figure 1—figure supplement 3*).

### Frp1 and Frp2 are required for heme-iron utilization

To directly examine the role of Frp1 and Frp2 in heme-iron acquisition, their genes were deleted and the hemoglobin-iron utilization phenotypes were analyzed by growth on YPD plates supplemented

with 1 mM bathophenanthroline disulfonate (BPS), an iron chelator, and hemoglobin as iron source. As shown in *Figure 1A*, the *frp1⁻/⁻* strain was unable to form colonies on the plate with hemoglobin as the sole iron source. Reintegration of a wild-type *FRP1* allele completely reverted the growth defect. In contrast, the *frp2⁻/⁻* strain showed no growth defect. We next performed this assay in medium buffered to pH 8.5. On the alkaline plates, we found that the *frp1⁻/⁻* strain was still unable to form colonies, whereas the *frp2⁻/⁻* strain showed a partial defect in heme-iron utilization, and reintegration of the *FRP2* wild-type allele complemented this growth defect (*Figure 1A*).

The heme-iron utilization was also tested in liquid medium, using a set of *frp1⁻/⁻*, *frp2⁻/⁻*, and *frp1⁻/⁻ frp2⁻/⁻* double deletion strains introduced in a background made defective in high-affinity iron acquisition by deletion of the copper transporter *CCC2* (*Weissman et al., 2002*), which makes the cells unable to grow in the presence of the iron chelator ferrozine. Growth of the mutants was assayed in the presence of increasing concentrations of hemoglobin. In unbuffered YPD (pH ~ 6.5), the same picture was seen as on plates, with the *frp1⁻/⁻* mutant unable to utilize hemoglobin-iron and the *frp2⁻/⁻* mutant unaffected (*Figure 1B*). In alkaline YPD (pH 8.5), the *frp1⁻/⁻* mutant failed to grow, whereas the *frp2⁻/⁻* mutant exhibited reduced growth, suggesting that this mutant is able to utilize hemoglobin, but at a reduced rate (*Figure 1B*). Reintegration of the wild-type *FRP1* and *FRP2* genes in the respective mutant backgrounds restored wild-type growth in unbuffered YPD, but did not completely restore growth at pH 8.5. This could be due to haploinsufficiency or to reduced expression of the reintegrated allele under these conditions.

We also compared growth of the *frp1⁻/⁻* and *frp2⁻/⁻* mutants to that of the CFEM protein mutants *rbt5⁻/⁻* and *pga7⁻/⁻*, on either hemoglobin or hemin as iron source. As shown in *Figure 1C*, in all media the *frp1⁻/⁻* mutant was as defective as the *pga7⁻/⁻* mutant, which is lacking the most essential CFEM protein (*Kuznets et al., 2014*). In YPD, both the *frp1⁻/⁻* and *pga7⁻/⁻* strains were unable to grow on hemoglobin, and the *rbt5⁻/⁻* mutant showed an intermediate phenotype, as noted before (*Kuznets et al., 2014*; *Weissman and Kornitzer, 2004*) whereas on hemin, growth of *frp1⁻/⁻* and *pga7⁻/⁻* was partially restored at higher concentrations, to similar extents. At pH 8.5, all mutants exhibited reduced growth on both hemoglobin and hemin, with *frp1⁻/⁻* and *pga7⁻/⁻* again showing the deepest defect (*Figure 1C*).

Since compared to the established FREs, Frp1 and Frp2 are relatively similar to each other, we asked whether differential expression of *FRP2* was the reason that it could not replace *FRP1*, that is, that the *frp1⁻/⁻* strain is unable to utilize hemoglobin even though *FRP2* is present in these cells. To answer this, we placed the *FRP2* open reading frame under the control of the *FRP1* promoter and transformed the resulting plasmid into the *frp1⁻/⁻* and *frp2⁻/⁻* strains. The *FRP1p-FRP2* construct could complement the *frp2⁻/⁻* but not the *frp1⁻/⁻* mutant (*Figure 1—figure supplement 4*), indicating that differential expression is not the reason that Frp2 is inactive in hemoglobin-iron utilization in unbuffered medium. Rather, this suggests that the Frp1 and Frp2 proteins are functionally distinct.

## Frp1 and Frp2 mediate heme uptake into the cytoplasm

To monitor heme uptake by the *frp1* and *frp2* mutants more directly, we took advantage of a recently developed cytoplasmic heme sensor system, based on a heme-quenchable GFP-cytochrome fusion. In this detection system, an mKATE2 red fluorescent domain linked to the GFP-cytochrome fusion serves as an internal fluorescence control. The ratio of GFP to mKATE2 fluorescence gives a measure of available cytoplasmic heme, with lower ratios indicating stronger GFP quenching due to higher cytochrome occupancy, that is, higher cytoplasmic heme concentrations. To expand the range of testable concentrations, a lower-affinity cytochrome domain mutant (M7A) can be used alongside the high-affinity original sensor (HS1) (*Hanna et al., 2016*; *Weissman et al., 2021*).

Wild-type, *frp1⁻/⁻*, *frp2⁻/⁻*, and *frp1⁻/⁻ frp2⁻/⁻* double mutant cells expressing the HS1 and M7A sensors were exposed to increasing hemin concentrations in the medium for 4 hr, and sensor fluorescence was measured. To ensure expression of the heme uptake system, the cells were grown with 1 mM ferrozine, which imposes iron limitation and activates the heme-uptake genes. The medium was buffered to pH 8.5, a condition where Frp2 was shown to be required as well (*Figure 1*). As can be seen in *Figure 2* (bottom panel), the low-affinity cytoplasmic sensor showed minimal occupancy in the absence of external hemin (fluorescence ratio of 10–12) and in the presence of hemin, only the wild-type cells showed a slight but significant increase in sensor occupancy. The wild-type HS1 sensor, in contrast, showed a fluorescence ratio of 2.5–3 in all strains in the absence of added hemin to the

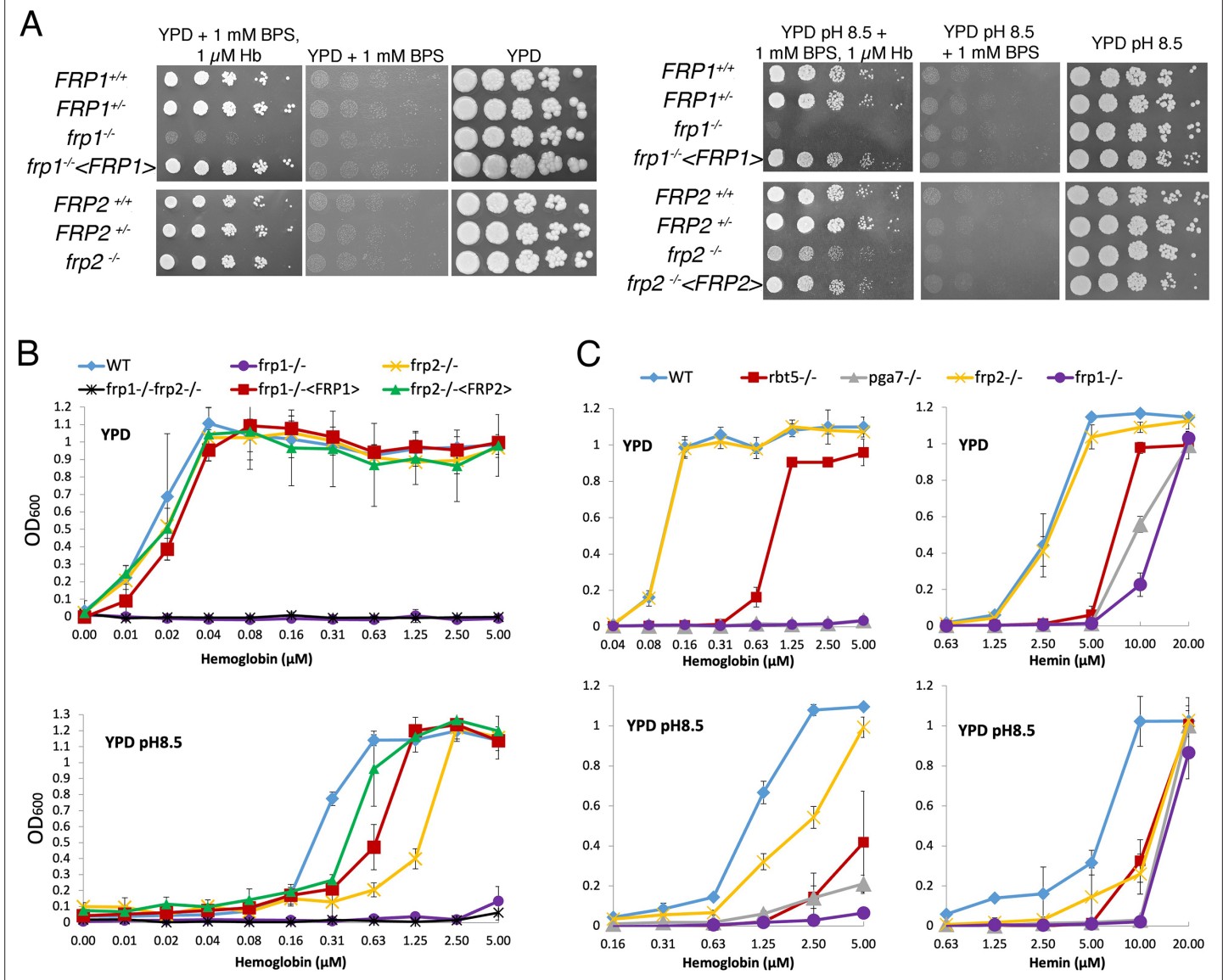

**Figure 1.** *FRP1* is essential for hemoglobin-iron acquisition, whereas *FRP2* contributes to growth on hemoglobin at alkaline pH. (**A**) Fivefold dilutions of cultures of the indicated strains were spotted on YPD or YPD pH 8.5, with the indicated supplements, and incubated for 3 days (Hb and BPS plates) or 2 days (YPD plates) at 30°C. WT = KC2, *FRP1+/+*=KC859, *frp1-/-*=KC870, *frp1-/-<FRP1 >* = KC1024, *FRP2+/+*=KC901, *frp2-/-*=KC912, *frp2-/-<FRP2 >* = KC1379. (**B**) The strains with indicated genotypes in the *ccc2-/-* background were grown in triplicates in YPD or YPD pH 8.5 media supplemented with 1 mM ferrozine and the indicated amounts of hemoglobin, and incubated at 30°C for 3 days. Each result is the average of three cultures. Standard deviations are indicated by vertical bars. WT = KC811, *frp1-/-*=KC1146, *frp2-/-*=KC1414, *frp1-/- frp2-/-*=KC1412, *frp1-/-<FRP1 >* = KC1146, *frp2-/-<FRP2>*=KC1411. (**C**) The *frp1-/-* and *frp2-/-* heme-iron utilization phenotype was compared to that of the CFEM protein mutants *rbt5-/-* and *pga7-/-*. The strains were grown in YPD or YPD pH 8.5, with 1 mM ferrozine and the indicated concentrations of hemoglobin or hemin, and grown and measured as in B. Wild type = KC68, *rbt5-/-*=KC139, *pga7-/-*=KC485, *frp1-/-*=KC923, *frp2-/-*=KC913. All strains in B and C carry a deletion of the *CCC2* gene, which causes a defect in high-affinity iron import and prevents growth in the presence of ferrozine.

The online version of this article includes the following source data and figure supplement(s) for figure 1:

**Source data 1.** Excel file with data used to make *Figure 1B*.

**Source data 2.** Excel file with data used to make *Figure 1C*.

**Figure supplement 1.** Proximity tree of Ascomycete CFEM protein sequences.

**Figure supplement 2.** Similarity between the Frp1 and Frp2 sequences and the predicted *C. albicans* ferric reductases sequences.

*Figure 1 continued on next page*

*Figure 1 continued*

**Figure supplement 3.** Synteny of the *FRP1-PGA7* (left) and *CSA1-FRP2* (right) genomic regions across Saccharomycetales species listed in the Candida Gene Order Browser database (cgob3.ucd.ie; *Fitzpatrick et al., 2010*; *Maguire et al., 2013*).

**Figure supplement 4.** *FRP2* under the control of the *FRP1* promoter is unable to complement the *frp1*[-/-] mutant but does complement the *frp2*[-/-] mutant.

medium, indicating a partial occupancy which reflects the steady-state free heme concentration in the cytoplasm. In medium supplemented with 10 µM hemin, in wild-type cells sensor occupancy became nearly maximal, indicating an increase in cytoplasmic hemin concentration. The increase in occupancy was much less in the *frp1*[-/-] mutant, and lesser still in the *frp2*[-/-] and *frp1*[-/-] *frp2*[-/-] strains, even when exposed to 30 and 50 µM hemin in the medium. Thus, while cellular heme concentrations do rise with extracellular heme, the increase in intracellular heme is much reduced in the *frp1*[-/-] and *frp2*[-/-] strains,

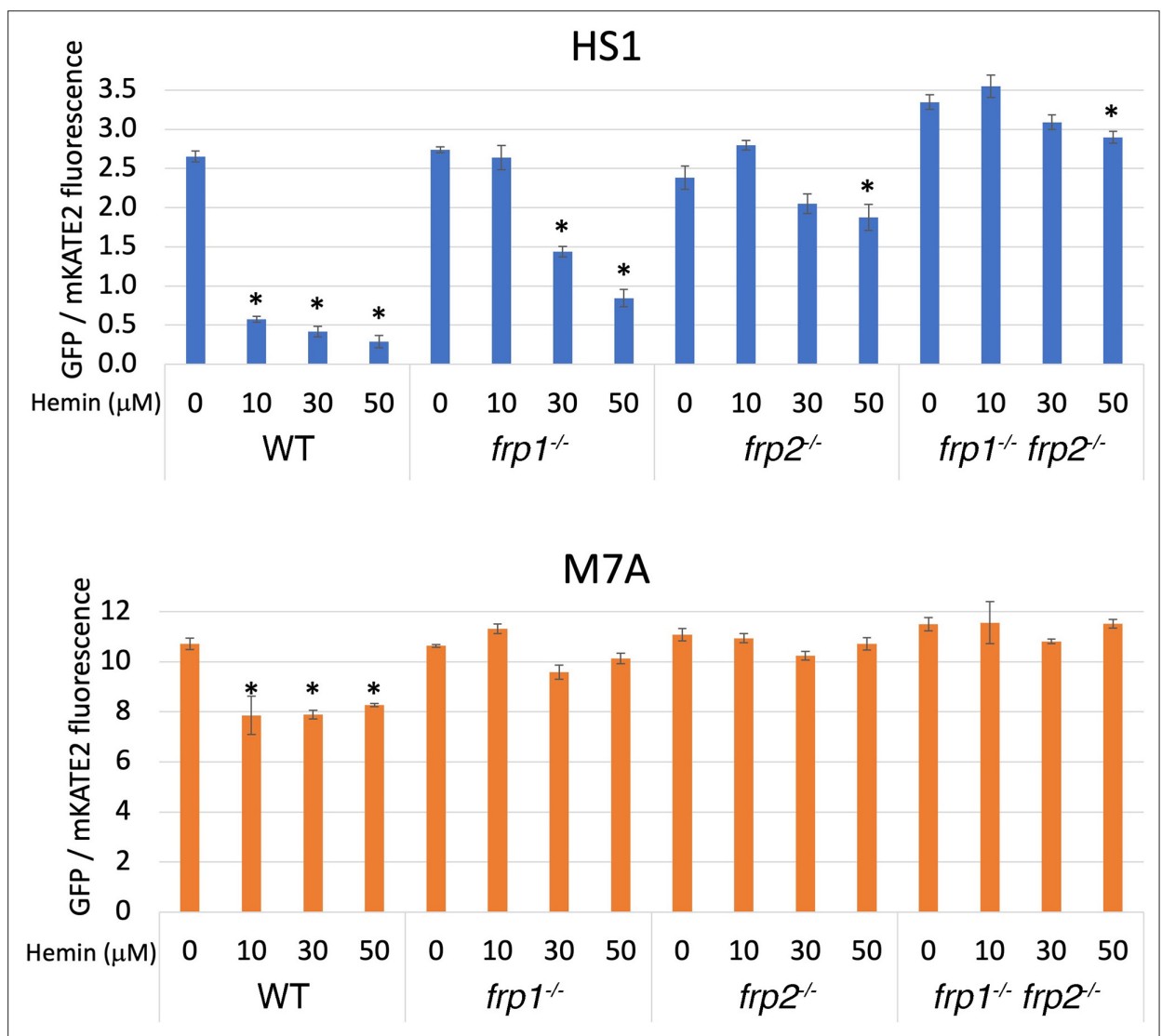

**Figure 2.** The *frp1*[-/-] and *frp2*[-/-] mutants are defective in heme uptake from the cytoplasm. The high-affinity HS1 and low-affinity M7A ratiometric heme sensors were used to monitor heme influx into the cytoplasm in the wild-type (KC2), *frp1*[-/-](KC870), *frp2*[-/-](KC912), and *frp1*[-/-] *frp2*[-/-] strains (KC1410) grown for 4 hr in YPD medium at pH 8.5 with 1 mM ferrozine, and with the indicated concentrations of hemin chloride. Each data point is the average of three different cultures, each measured twice. Vertical bars indicate standard deviations, and the asterisks indicate measurements that are significantly different from the 0 µM hemin reading with p≤0.0001.

The online version of this article includes the following source data for figure 2:

**Source data 1.** Excel file with data used to make *Figure 2C*.

and almost inexistent in the *frp1⁻/⁻ frp2⁻/⁻* strain. The reduced increase in sensor occupancy in the *frp2⁻/⁻* mutant exposed to external hemin compared to the *frp1⁻/⁻* mutant suggests that Frp2 is more important for heme influx into the cytoplasm than Frp1 under these conditions, in spite of Frp1 being dominant for heme utilization even at pH 8.5 (*Figure 1*).

## *FRP1* and *FRP2* are differentially required for non-iron metalloprotoporphyrin uptake and toxicity

The non-iron metalloprotoporphyrins (MPPs) are synthetic heme derivatives that have a non-iron metal atom instead of the iron at the center of the porphyrin ring structure. Many MPPs are toxic to bacteria, and this toxicity depends on the bacterial heme-import pathways (*Hijazi et al., 2017*; *Mitra et al., 2017*; *Olczak et al., 2012*; *Stojiljkovic et al., 2001*; *Wakeman et al., 2014*). Two MPPs, Ga³⁺-protoporphyrin IX (Ga-PPIX) and Mn³⁺-protoporphyrin IX (Mn-PPIX) were also found to be toxic to the pathogenic fungus *C. neoformans*, and their toxicity depended on some of the same factors required for heme uptake in that organism (*Bairwa et al., 2019*; *Hu et al., 2015*). We tested toxicity of these two MPPs as well as of Co³⁺-protoporphyrin IX (Co-PPIX) and Zn²⁺-protoporphyrin IX (Zn-PPIX) in YPD medium supplemented with 1 mM ferrozine to induce the heme-uptake pathway, and found that all of them are toxic to *C. albicans*. To investigate the roles of Frp1 and Frp2 in uptake of these compounds, we then tested the *frp1⁻/⁻, frp2⁻/⁻*, and double mutants for Ga-PPIX, Mn-PPIX, Co-PPIX, and Zn-PPIX sensitivity in regular YPD medium and in YPD medium buffered to pH 7.5 (*Figure 3A*). In regular YPD medium, the *frp1⁻/⁻* mutant showed increased resistance to all of the MPPs tested except Ga-PPIX. The *frp2⁻/⁻* mutant showed increased resistance to Zn-PPIX and to a lesser extent to Mn-PPIX but not to Co-PPIX or to Ga-PPIX. At pH 7.5, the *frp2⁻/⁻* strain but not the *frp1⁻/⁻* strain showed increased resistance to Ga-PPIX. The *frp2⁻/⁻* strain also showed increased resistance to Mn-PPIX and Zn-PPIX, and to a lesser extent, so did the *frp1⁻/⁻* strain (Co-PPIX was not toxic at pH 7.5). In summary, Frp1 and Frp2 were required for full toxicity of at least some of the MPPs in either unbuffered YPD medium (pH ~ 6.5) or in medium buffered to pH 7.5, with Frp1 being more important for toxicity in unbuffered medium and Frp2 for toxicity at pH 7.5.

In view of the essential role of the CFEM hemophore Pga7 in heme acquisition, and of the fact that CFEM hemophores can bind GaPPIX, MnPPIX, and to some extent CoPPIX (our unpublished data), we tested whether Pga7 is required for MPP toxicity. As shown in *Figure 3—figure supplement 1*, no difference could be detected in the sensitivity of *pga7⁻/⁻* vs. *PGA7* cells, indicating that Pga7 is not involved in the uptake of these compounds.

Next, we checked if Frp1 and Frp2 can mediate uptake of Zn²⁺-mesoporphyrin (ZnMP). ZnMP is a fluorescent porphyrin that structurally resembles heme and that has been used to analyze heme uptake pathways in mammalian and fungal cells (*Mourer et al., 2017*; *Worthington et al., 2001*). We found that in a wild-type strain, iron starvation induces the ability of the cells to take up ZnMP, and the addition of hemin to the medium efficiently inhibits it, suggesting that ZnMP is taken up by the same pathway as heme (*Figure 3—figure supplement 2*). We also noted that unlike in *S. pombe* and mammalian cells, where the ZnMP fluorescent signal is diffusely cytoplasmic, in *C. albicans*, the fluorescent signal remains mainly associated with membrane-like structures.

Since in cells growing in regular iron-replete medium, ZnMP uptake is negligible, presumably due to lack of expression of the heme-uptake system genes, we could ask whether Frp1 or Frp2 alone were sufficient to mediate ZnMP uptake by ectopically expressing them under these conditions. *FRP1* and *FRP2* were placed under the promoter of *SSB1*, a gene that is strongly expressed under different growth conditions (*Ofir et al., 2012*), and ZnMP uptake was monitored in these cells. Both *SSB1p-FRP1*- and *SSB1p-FRP2*-harboring cells showed a strong intracellular ZnMP signal (*Figure 3B*). This indicates that Frp1 and Frp2 are each sufficient to mediate uptake of ZnMP.

## Frp1 and Frp2 relocalize to the cell surface in the presence of heme

FREs are typically located on the plasma membrane, where they reduce extracellular iron (*Lesuisse et al., 1987*; *Shatwell et al., 1996*; *Yun et al., 2001*). To determine the subcellular localization of Frp1 and Frp2, we constructed an Frp1-GFP and an Frp2-GFP fusion under the control of their endogenous promoters. These fusion proteins retained heme uptake activity, as evidenced by their ability to support growth on hemoglobin (*Figure 4—figure supplement 1*). Both the Frp1-GFP and Frp2-GFP signals became visible after induction by iron starvation and were localized at the cell surface and at

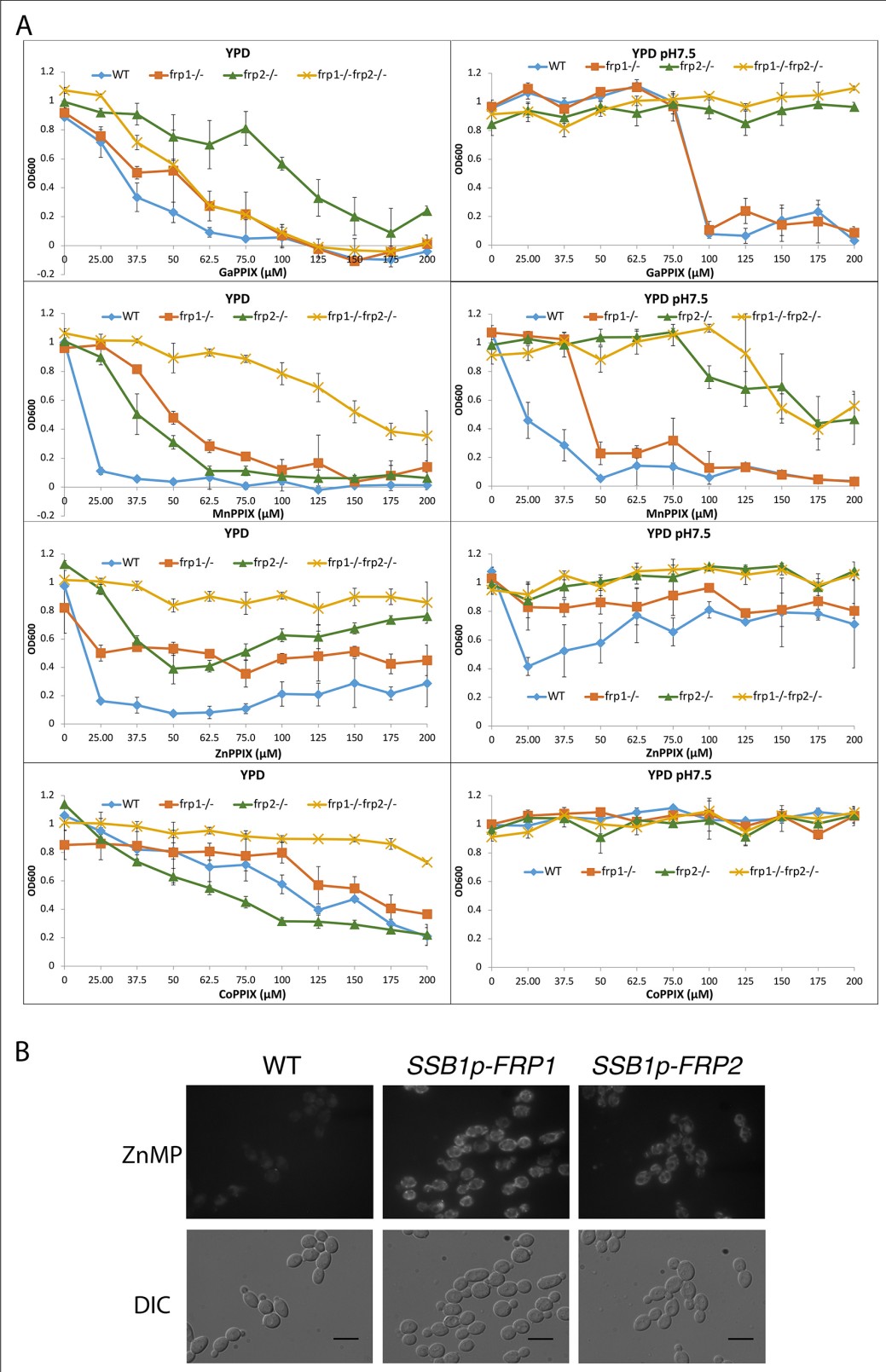

**Figure 3.** Frp1 and Frp2 participate in the uptake of heme homologs. (**A**) *FRP1* and *FRP2* are differentially required for sensitivity to toxic heme homologs. The indicated strains were diluted in YPD medium with different concentrations of metal-protoporphyrin IX compounds, as indicated, and grown in 96-well plates at 30°C for 2 days. The graph points indicate the averages of triplicate cultures, and the standard deviations are indicated by

*Figure 3 continued on next page*

*Figure 3 continued*

vertical bars. The strains used are KC590 (WT), KC966 (*frp1⁻ᐟ*), KC1053 (*frp2⁻ᐟ*), KC1061 (*frp1⁻ᐟ frp2⁻ᐟ*). (**B**) Expression of either *FRP1* or *FRP2* is sufficient to enable ZnMP uptake. Wild-type strain KC2 (WT), strain KC1080 that has a single *FRP1* gene under the *SSB1* promoter (*SSB1p-FRP1*) and strain KC1244 that has a single *FRP2* gene under the *SSB1* promoter (*SSB1p-FRP2*) were grown in YPD to log phase, then exposed to 1 mM ZnMP for 10 min, washed and visualized by epifluorescence microscopy. Scale bar = 5 μm.

The online version of this article includes the following source data and figure supplement(s) for figure 3:

**Source data 1.** Excel file with data used to make *Figure 3A*.

**Figure supplement 1.** Pga7 is not required for sensitivity to non-iron metalloprotoporphyrins (MPPs).

**Figure supplement 2.** Hemin competes with zinc-mesoporphyrin for uptake by *Candida albicans* cells.

---

the endoplasmic reticulum (ER), and the GFP signals also accumulated in the vacuole (*Figure 4A*). Identification of the vacuole and perinuclear ER membrane was verified with CMAC, a fluorophore specific for the vacuole and Hoechst 33342, a nuclear stain, respectively (*Figure 4A*).

Induction of Frp1-GFP and Frp2-GFP was monitored by shifting log-phase cells growing in YPD to YPD+ferrozine (Frp1-GFP) or to YPD pH 8.5+ferrozine (Frp2-GFP), without or with added hemin. Samples were removed 1, 2, and 3 hr after induction for microscopic observation and for protein analysis by Western blot. By microscopic observation, the GFP signal increased with time, and in the absence of hemin, was mostly detected in the vacuole. In the presence of hemin, the plasma membrane signal became much more prominent (*Figure 4B*). In addition, Frp2-GFP expression appeared higher, and more prominently vacuolar, compared to Frp1-GFP. Analysis of the proteins by Western blot also showed induction of Frp1-GFP and Frp2-GFP in iron-limited medium (*Figure 4—figure supplement 2A*), and showed decay of the protein upon shift back to YPD medium (*Figure 4—figure supplement 2B*). This analysis furthermore indicated that Frp2-GFP expression is higher than Frp1-GFP. Under all conditions, only full-length GFP fusion proteins were detectable, suggesting that the GFP signals visible on the micrographs represent full-length, potentially active protein.

In order to measure the kinetics of relocalization of Frp1-GFP and Frp2-GFP to the cell surface in the presence of hemin, cells induced as above for 3 hr were exposed to hemin, and the GFP signal localization was monitored every 10 min. A significant increase in cell surface GFP signal was detectable for both proteins after 30 min exposure to hemin (*Figure 4C*).

## Expression of Frp1 in *S. cerevisiae* promotes heme uptake in the presence of Pga7

The data shown above indicate that Frp1 and Frp2 are able to mediate the uptake of metal-substituted heme analogs even in the absence of the extracellular hemophore cascade, represented by the membrane-bound hemophore Pga7 (*Figure 3A*, *Figure 3—figure supplement 1*). The CFEM hemophores are however necessary for heme uptake. To better characterize the minimal requirements for Frp1-mediated heme uptake, we attempted to reconstitute it in *S. cerevisiae*, an organism lacking an efficient heme uptake system, using *FRP1* expressed under the strong H2B histone gene promoter. In order to assess the requirement of the extracellular hemophore cascade in Frp1-mediated heme uptake in *S. cerevisiae*, *FRP1* was expressed either in the presence or absence of *PGA7* under the adjacent H2A histone gene promoter. To monitor heme uptake, we used a *hem1Δ* mutant, unable to synthesize heme. The *S. cerevisiae hem1Δ* strain is usually maintained in the presence of the Hem1 product δ-aminolevulinic acid (ALA) in the medium, however it can also grow when supplied with a high enough hemin concentrations in the medium.

As shown in *Figure 5A*, expression of Frp1 together with Pga7 promoted better growth of the *hem1* strain on plates containing hemin. In liquid medium, expression of Frp1 together with Pga7 enabled growth of the *hem1* strain at much lower hemin concentrations than in their absence, consistent with a more efficient heme uptake in these cells (*Figure 5B*, 2 days). Upon longer incubation, expression of Frp1 alone, but not of Pga7 alone, was found to promote some growth over background as well (*Figure 5B*, 3 days). This observation indicates that Frp1 by itself can promote heme uptake into the cell, albeit with lower efficiency.

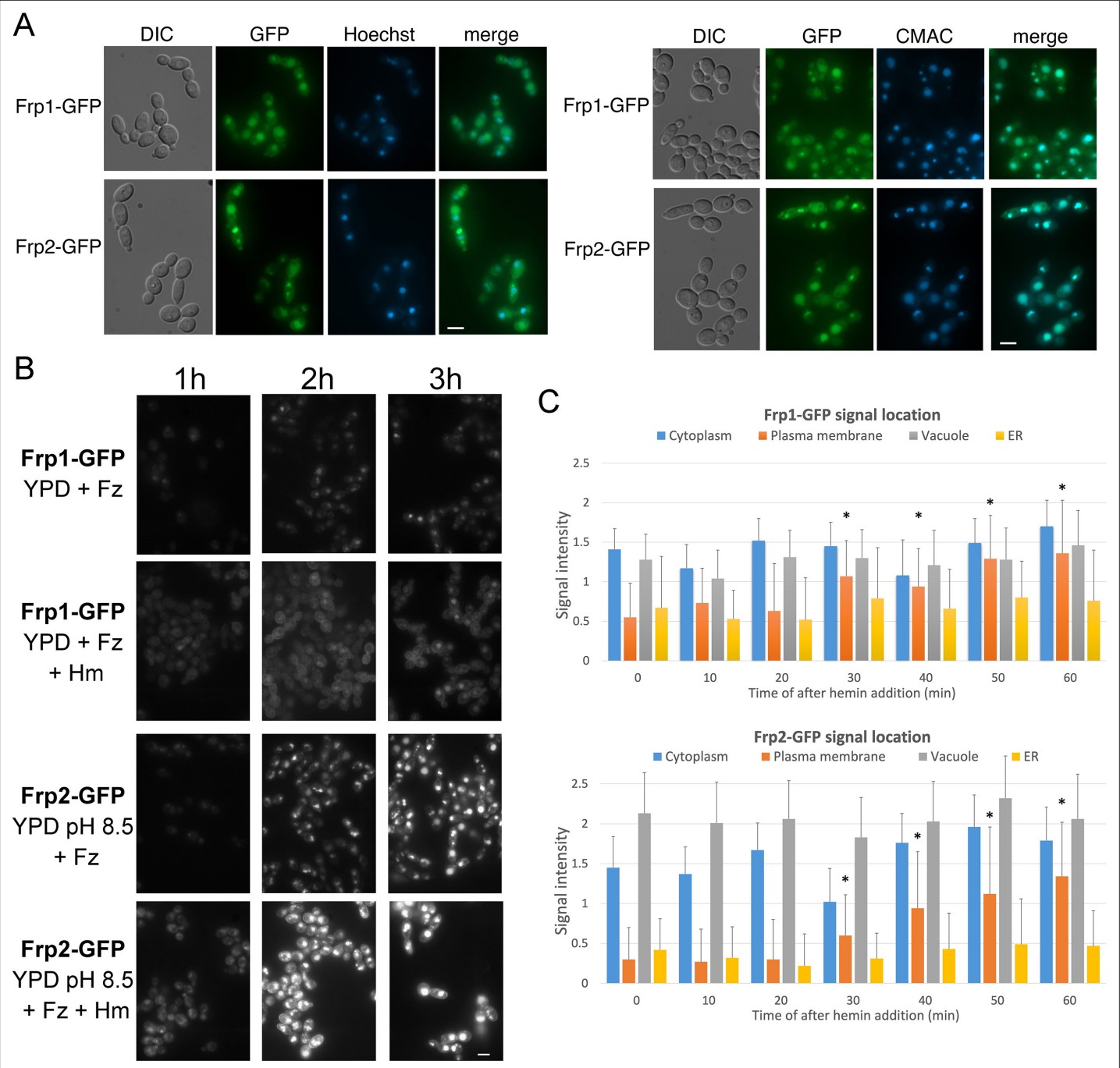

**Figure 4.** Subcellular localization of Frp1-GFP and Frp2-GFP fusion proteins. (**A**) The cells (Frp1-GFP=KC914, Frp2-GFP=KC1405) were grown in iron-limited medium for 3 hr. Left panels: Localization of the Frp-GFP proteins vs. the nuclear stain Hoechst 33324. Right panels: Localization of the Frp-GFP proteins vs. the vacuole stain CMAC. Scale bars = 5 µm. (**B**) Location of Frp1-GFP and Frp2-GFP after induction by iron starvation, without and with added 50 µM hemin. The cells were grown to late-log phase in YPD, then shifted to the indicated media, and visualized at the indicated times by epifluorescence microscopy. Scale bar = 5 µm. (**C**) Kinetics of Frp1/2-GFP relocation after exposure to hemin. The cells were grown in iron-limited medium for 3 hr and then 50 µM hemin was added. The graphs describe quantitation of subcellular localization of the Frp1-GFP and Frp2-GFP signals after exposure to hemin. At least 100 cells were observed for each timepoint, and the signal intensity at each subcellular location was assigned a value from 0 to 3. The graph indicates the average intensities at each of four cellular locations. Note that 'ER' denotes location on the perinuclear membrane and its projections, whereas 'plasma membrane' could possibly also include cortical ER, which cannot be differentiated at this level of resolution. The asterisks indicate the plasma membrane values that differ statistically from t=0' with p<0.00001 by Mann-Whitney's U test.

The online version of this article includes the following source data and figure supplement(s) for figure 4:

*Figure 4 continued on next page*

*Figure 4 continued*

**Source data 1.** Excel file with data used to make *Figure 4C*.

**Figure supplement 1.** FRP1-GFP and FRP2-GFP fusion proteins retain heme uptake activity and support growth on hemoglobin.

**Figure supplement 2.** Only full-length Frp1-GFP and Frp2-GFP proteins are detectable under all conditions.

**Figure supplement 2—source data 1.** Original Western blots used to make *Figure 4—figure supplement 2 A-B*.

## Homology between CFEM proteins and the FRE N-terminal domains

The structure of Csa2, a CFEM hemophore with high sequence homology to Pga7, indicated, at the time when it was solved, that it assumes a new protein fold, lacking any structural homologs in the 3D protein structure database (*Nasser et al., 2016*). We took advantage of the recent availability, through the Alphafold project (*Jumper et al., 2021*), of high-quality structure predictions of all *C. albicans* proteins, to search again for CFEM structural homologs. Remarkably, beyond the five related CFEM hemophore proteins, six out of the eight closest structural homologs to Csa2 were *C. albicans* FREs (*Supplementary file 1*). Alignment of the regions of homology showed that they correspond to the predicted extracellular N-termini of these FREs (*Figure 6—figure supplement 1*). Despite the low sequence homology between these proteins and Csa2, six of the eight canonical CFEM cysteine residues, forming three out of the four disulfide bonds in the CFEM structure, are nonetheless conserved and similarly spaced in the predicted FRE N-termini, and indeed are predicted to form disulfide bonds in the predicted structures (*Figure 6—figure supplement 1*). We next extended the sequence comparison to all other *C. albicans* FREs. Of the 12 FRE homologs that have an extended N-terminal domain (*Figure 1—figure supplement 2*), 10 have six cysteines with conserved CFEM alignment, whereas two have only four of the six cysteines conserved (*Figure 6*). An additional key residue of the CFEM domain, the heme iron-coordinating aspartic acid (*Nasser et al., 2016*), is conserved in four FREs and is replaced by asparagine in six others. Taken together, this analysis suggests that the substrate-binding N-terminal domain of fungal FREs assumes the CFEM fold.

## Discussion

*C. albicans* can utilize external heme as both an iron source and a heme source, using an uptake system that depends on a group of related hemophores that contain a unique CFEM-type heme-binding domain (*Moors et al., 1992*; *Nasser et al., 2016*; *Weissman et al., 2021*). This pathway involves capture of the heme from host proteins by the CFEM hemophores, and its transfer from one hemophore to the next across the cell wall to the cell membrane (*Kornitzer and Roy, 2020*; *Kuznets et al., 2014*; *Pinsky et al., 2020*; *Figure 7*). While vacuolar function and the endocytic pathway were found to be essential for heme-iron utilization (*Weissman et al., 2008*), the mechanism for heme internalization into the cell was unknown. Here, we fill this gap, and identify a family of membrane proteins, previously classified as FREs based on sequence homology (*Almeida et al., 2009*; *Baek et al., 2008*), that are essential for heme uptake into the cell.

FRP1 and *FRP2* have been previously identified as genes that are highly induced under iron starvation conditions by the Hap43/Cap2-Sfu1 system, as well as under alkaline conditions by the Rim101 transcription factor (*Baek et al., 2008*; *Bensen et al., 2004*; *Chen et al., 2011*; *Lan et al., 2004*; *Liang et al., 2009*; *Singh et al., 2011*). Indeed, in some reports, *FRP1* is the highest-induced gene under iron starvation conditions, together with the CFEM hemophore gene *RBT5* (*Lan et al., 2004*). Here, we identified *FRP1* and *FRP2* by phylogenetic profiling as genes that co-segregate with CFEM hemophore genes across fungal genomes. The function of these genes was confirmed experimentally: we showed here that *FRP1* is essential for heme uptake under all conditions tested, and that *FRP2* contributes to heme uptake under alkaline conditions. Interestingly, a predicted FRE, CnFre2/CNAG_06821, was identified in the distantly related basidiomycete fungal pathogen *C. neoformans*, as being required for optimal growth on hemin as well (*Saikia et al., 2014*). However, unlike Frp1/2, CnFre2 is not required for sensitivity to toxic heme analogs (*Saikia et al., 2014*), and CnFre2's N-terminal domain lacks the cysteine arrangement found in CFEM proteins and in *C. albicans* FRE N-terminal domains. It is thus unclear at this point whether CnFre2 and Frp1/2 share a common mechanism.

*FRP1* and *FRP2* are adjacent to, and share a promoter region with, the two CFEM protein genes *PGA7* and *CSA1*, respectively. This synteny is maintained across most fungal genomes analyzed, further

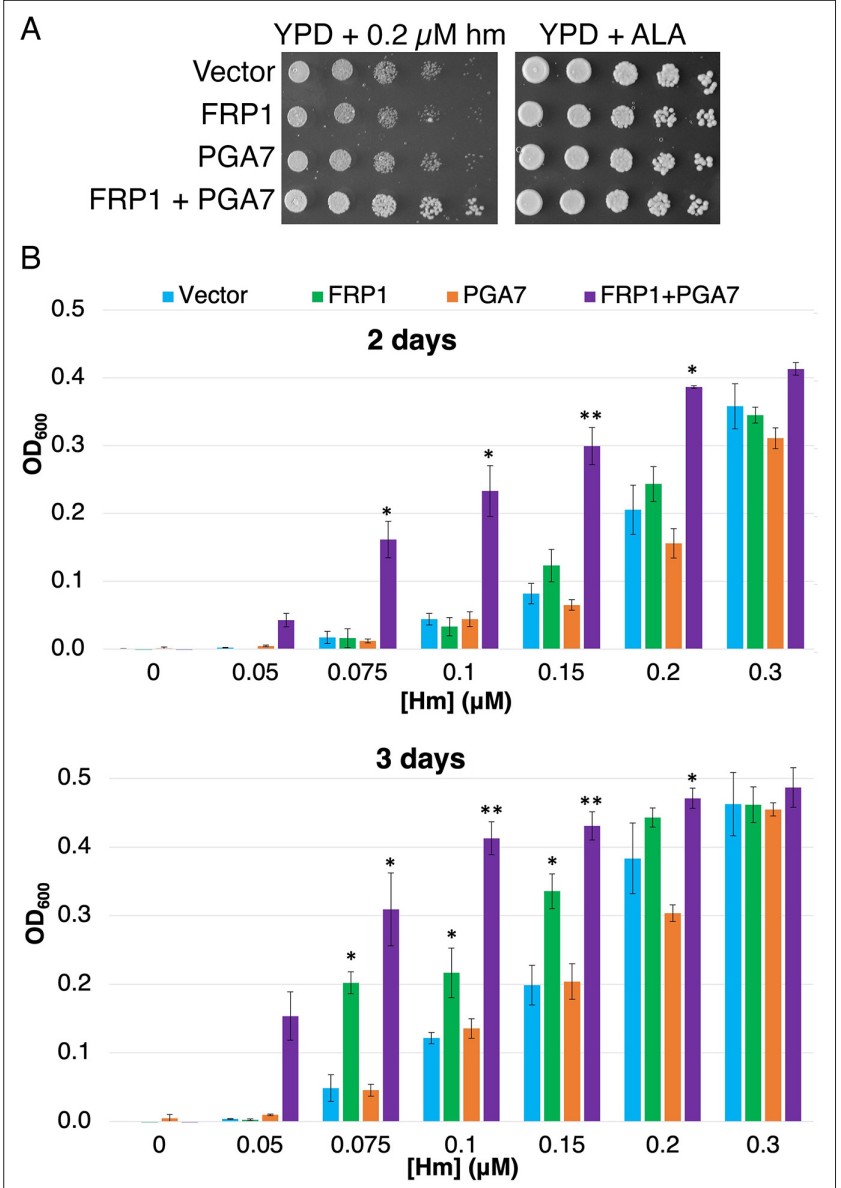

**Figure 5.** Frp1 and Pga7 collaborate in heme uptake. (**A**) *Saccharomyces cerevisiae hem1Δ* cells (KY1498) were transformed with a vector plasmid, or with plasmids *HTB2p-FRP1* (KB2569), *HTA2p-PGA7*(KB2789) or *HTB2p-FRP1 HTA2p-PGA7* (KB2566) and drop-diluted on SC-HIS plates supplemented with either 0.2 μM hemin or 50 μg/ml δ-aminolevulinic acid (ALA), as indicated. The plates were incubated for 2 days at 30°C. (**B**) The same strains were diluted in SC-HIS medium supplemented with the indicated amounts of hemin, and incubated at 30°C for 2 or 3 days, as indicated. For each plasmid, three independent transformant colonies were grown. The data indicate the average of the three cultures, and the error bars indicate the standard deviations. Statistically significant differences compared to vector control are indicated with one asterisk (p<0.05) or two asterisks (p<0.001) (Student's t-test).

The online version of this article includes the following source data for figure 5:

**Source data 1.** Excel file with data used to make *Figure 5B*.

suggesting a common function in heme acquisition. We note however that while *PGA7* was shown to be the main CFEM hemophore required for heme uptake under all conditions tested (**Kuznets et al., 2014**; **Nasser et al., 2016**), we have been unable so far to detect any phenotype in the *csa1[-/-]* deletion mutant, either alone or in combination with other CFEM gene mutants, in either regular or alkaline medium. Given the high homology of the Csa1 tandem CFEM domains to that of Rbt5 and other

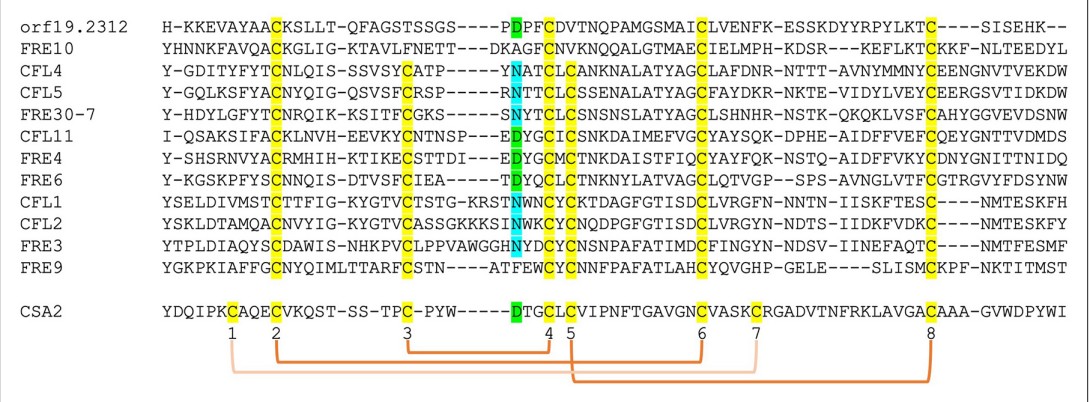

**Figure 6.** Alignment by MAFFT (*Katoh et al., 2005*) of the N-termini of 12 FRE-like proteins. The alignment with Csa2 was superimposed based on the structural alignment shown in *Figure 6—figure supplement 1*. The eight CFEM cysteines are highlighted in yellow and numbered below the sequence. The orange linkers connect between the cysteines that form disulfide bonds in the Csa2 structure. The Csa2 heme iron-coordinating Asp residue and its homologs in ferric reductases (FREs) are highlighted in green, and the corresponding Asn residues in blue.

The online version of this article includes the following figure supplement(s) for figure 6:

**Figure supplement 1.** Alignment between the Csa2 CFEM structure and predicted FRE N-termini.

**Figure supplement 2.** Alignment of the Alphafold-predicted structures of the ferric reductase Cfl11 (grey), of Frp1 (orange-red), and of the Pga7 CFEM domain (green).

established hemophores, it is nonetheless likely that Csa1 does play a role in heme acquisition under conditions yet to be identified.

Regarding *PGA7*, beyond synteny with *FRP1*, the similar phenotype of the *frp1⁻/⁻* and the *pga7⁻/⁻* mutants in *C. albicans* and the effect of co-expression of *FRP1* and *PGA7* in *S. cerevisiae* indicated a close cooperation between these proteins in the heme uptake pathway. Furthermore, structure and sequence comparison between FREs, Frp1/2 and CFEM hemophores showed that while the Frp1/2 sequences align with the transmembrane and intracellular portions of the FREs, the CFEM hemophores sequences such as Pga7 are related to the N-terminal extracellular domain of the FREs (*Figure 6—figure supplement 2*). This suggests that the Frp-Pga7 complex is evolutionarily related to the main body and the N-terminal domain, respectively, of the FRE protein family. The predicted CFEM fold of the various FRE N-termini could, by analogy with the CFEM heme-binding platform (*Nasser et al., 2016*), form a binding site for the various types of substrates of these FREs around the conserved Asp/Asn.

The known role of FREs is to reduce ferric iron in order to extract it from chelates and siderophores (*Philpott, 2006*; *Yun et al., 2000*). The homology of Frp1/2 to FREs therefore begs the question: why would an FRE activity be required for heme uptake? A possible answer comes from the observation that the CFEM domain specifically binds $Fe^{+3}$-heme (ferriheme), but not $Fe^{+2}$-heme (ferroheme), thanks to its unusual Asp-mediated iron coordination (*Nasser et al., 2016*). This redox sensitivity of binding appears to be essential for the CFEM hemophore function, because mutant proteins carrying a substitution of the iron-coordination Asp with His can efficiently bind ferroheme as well as ferriheme, but these Asp-to-His mutants are completely inactive in vivo (*Nasser et al., 2016*). The homology of Frp1/2 to FREs and their role in heme acquisition therefore raise the possibility that they function as heme reductases, and that the Frp1/2 proteins reduce heme bound to the extracellular CFEM hemophores in order to release it.

If Frp1 and Frp2's role is to release heme from Pga7 or other CFEM hemophores, what then is the protein that binds the released heme and mediates its internalization into the cell? An important clue for answering this question is the observation that Frp1 and Frp2 – but not Pga7 – mediate the uptake of metal-substituted heme analogs such as fluorescent ZnMP and the toxic metal-substituted PPIX derivatives. These metal-substituted analogs are either not at all redox-active, or much less so compared to heme, indicating that heme reduction cannot be the sole function of Frp1/2. Rather, these observations suggest that Frp1/2 function in the transport of these molecules into the cell as well. The location of Frp1/2 in the vacuole and their partial relocalization in the presence of hemin is

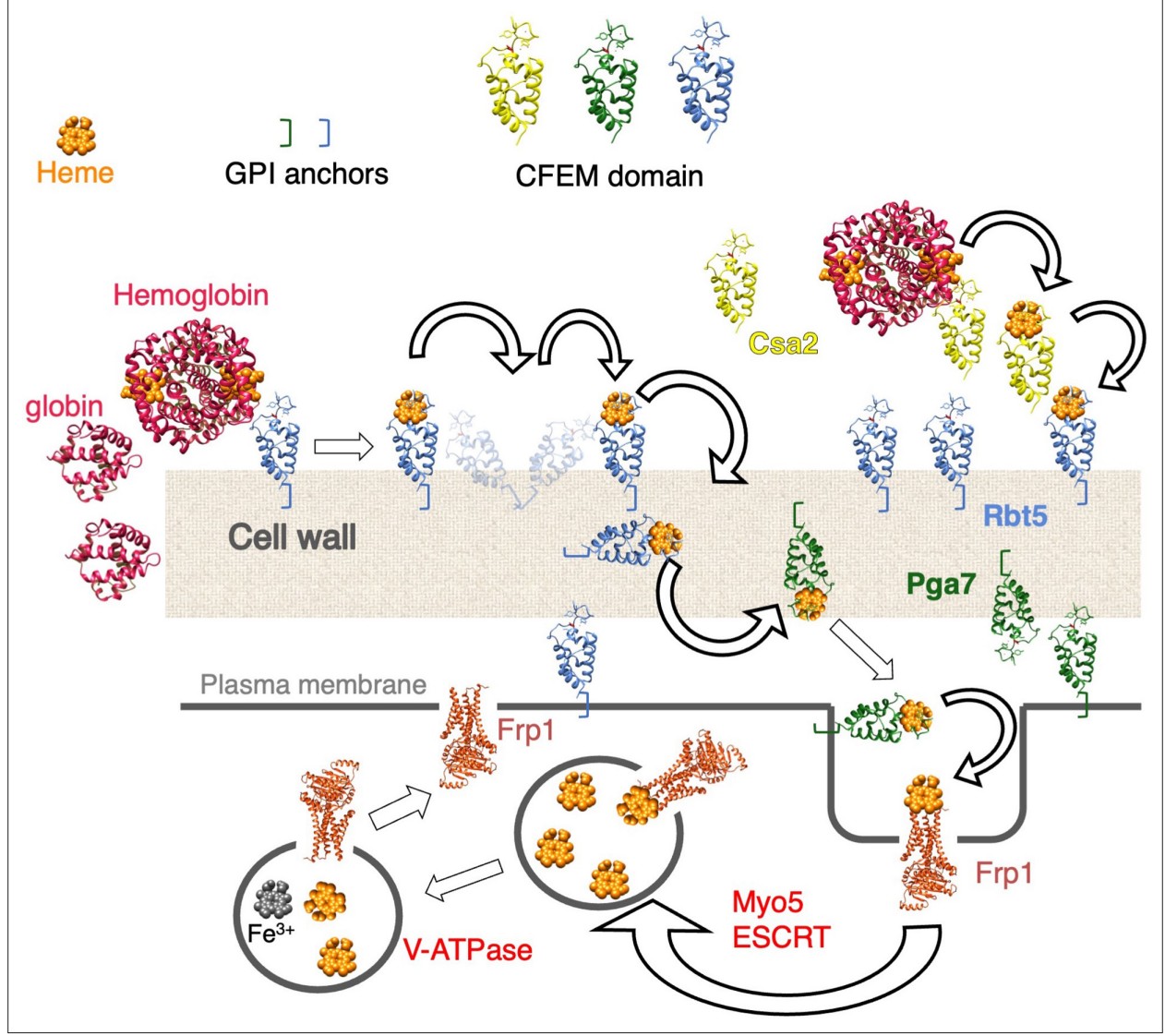

**Figure 7.** Proposed model for heme uptake in *Candida albicans*. See Discussion for details.

consistent with a role of Frp1/2 as heme receptors shuttling between the plasma membrane and the vacuole (*Figure 7*).

An intriguing question is the reason for the existence of two distinct paralogs, Frp1 and Frp2, in *C. albicans* as well as in many fungal genomes containing FRP homologs. Whereas *C. albicans* Frp1 is essential for heme uptake under all the conditions we tested, Frp2 contributed to heme-iron utilization under alkaline conditions only. With regard to the roles of Frp1 and Frp2 in sensitivity to metal-substituted heme analogs, *FRP1* was the main determinant of sensitivity in unbuffered YPD medium (pH ~ 6.5), except for GaPPIX where *FRP2* played the dominant role, and *FRP2* also played a role in ZnPPIX and MnPPIX sensitivity. At neutral-alkaline pH, the Frp2 contribution became more important yet. While differential expression of *FRP1* and *FRP2* might be invoked to explain some of these results, it does not explain differential phenotypes with different metal-substituted heme analogs under the same conditions. Furthermore, expression of *FRP2* under the *FRP1* promoter could not complement the *frp1⁻/⁻* mutant phenotype for heme uptake, indicating that the two proteins do have distinct functions. Localization of the GFP fusions of Frp1 and Frp2 indicated a similar distribution between vacuole and cell surface, with however a higher apparent concentration of Frp2 in the vacuole (*Figure 4B and C*). One possibility, relating to the Frp proteins' activity as reductases, is that, since liganded iron was reported to be more readily reduced at more acidic pH (*Dhungana and Crumbliss, 2005*), combined

activity of Frp1 and Frp2 might be required to reduce Pga7-bound heme at neutral and alkaline pH. Another possibility, relating to the Frp proteins' function as heme receptors, is that the affinity of heme for Frp1 and Frp2 differs depending on pH. Differential requirement of Frp1 and Frp2 for the toxicity of specific metal-substituted protoporphyrins could similarly reflect differential affinity of these compounds to Frp1 and Frp2. A third possibility, which takes into account that Frp2 showed a higher relative concentration in the vacuole compared to Frp1 (*Figure 4*), is that Frp2 specifically plays a role, distinct from that of Frp1, in subcellular distribution of heme. Distinct roles for Frp1 and Frp2 in redistribution of imported heme between the vacuole and other compartments, and differential effects of pH on this process, might explain why Frp1 is functionally more important for heme-iron utilization even at alkaline pH (*Figure 1*) while Frp2 is more active in promoting heme influx into the cytoplasm (*Figure 2*).

To conclude, we have identified two FRE-related membrane proteins, Frp1 and Frp2, as the 'missing link' between the extracellular CFEM hemophore cascade and the intracellular heme endocytosis pathway to the vacuole. Our data suggest that these proteins could function both as heme reductases and as heme receptors that bring the heme from the cell surface into the cell (*Figure 7*). The Frp1/2 -CFEM hemophore pathway thus represent a new paradigm for heme uptake and utilization.

# Materials and methods

**Key resources table**

| Reagent type (species) or resource | Designation | Source or reference | Identifiers | Additional information |
|---|---|---|---|---|
| Strain, strain background (*Candida albicans*) | WT | *Fonzi and Irwin, 1993* | KC2=CAF3-1 | *ura3Δ::imm434/ura3Δ::imm434* |
| Strain, strain background (*Candida albicans*) | *ccc2⁻/⁻* | *Weissman et al., 2002* | KC68 | KC2 *ccc2Δ::hisG/ccc2Δ::hisG* |
| Strain, strain background (*Candida. albicans*) | *rbt5⁻/⁻* | *Weissman and Kornitzer, 2004* | KC139 | KC68 *rbt5Δ/rbt5Δ* |
| Strain, strain background (*Candida. albicans*) | *pga7⁻/⁻* | *Kuznets et al., 2014* | KC485 | KC68 *pga7Δ/pga7Δ* |
| Strain, strain background (*Candida albicans*) | WT | *Fonzi and Irwin, 1993* | KC590=CAI4 | *ura3Δ::imm434/ura3Δ::imm434* |
| Strain, strain background (*Candida albicans*) | *pga7⁻/⁻* | *Kuznets et al., 2014* | KC646 | KC590 *pga7Δ/pga7Δ ADE2/ade2::URA3* |
| Strain, strain background (*Candida. albicans*) | *pga7⁻/⁻ PGA7 URA3* | *Kuznets et al., 2014* | KC647 | KC590 *pga7Δ/pga7Δ ADE2/ade2::PGA7 URA3* |
| Strain, strain background (*Candida albicans*) | *pga7⁻/⁻ URA3* | This work | KC811 | KC68 *ADE2/ade2::URA3* |
| Strain, strain background (*Candida albicans*) | *FRP1⁺/⁻* | This work | KC859 | KC2 *FRP1/frp1Δ* |
| Strain, strain background (*Candida albicans*) | *frp1⁻/⁻* | This work | KC870 | KC2 *frp1Δ/frp1Δ* |
| Strain, strain background (*Candida albicans*) | *FRP1⁺/⁻* | This work | KC901 | KC2 *FRP2/frp2Δ* |

*Continued on next page*

*Continued*

| Reagent type (species) or resource | Designation | Source or reference | Identifiers | Additional information |
|---|---|---|---|---|
| Strain, strain background (*Candida albicans*) | *frp2⁻/⁻* | This work | KC912 | KC2 *frp2Δ/frp2Δ* |
| Strain, strain background (*Candida albicans*) | *frp2⁻/⁻* | This work | KC913 | KC68 *frp2Δ/frp2Δ* |
| Strain, strain background (*Candida albicans*) | *FRP1-GFP* | This work | KC914 | KC2 *FRP1/FRP1-GFP URA3* |
| Strain, strain background (*Candida albicans*) | *FRP1-GFP* | This work | KC916 | KC2 *frp1Δ/FRP1-GFP URA3* |
| Strain, strain background (*Candida albicans*) | *frp1⁻/⁻* | This work | KC923 | KC68 *frp1Δ/frp1Δ* |
| Strain, strain background (*Candida albicans*) | *frp1⁻/⁻* | This work | KC966 | KC590 *frp1Δ/frp1Δ* |
| Strain, strain background (*Candida albicans*) | *frp1⁻/⁻ URA3* | This work | KC1023 | KC870 *ADE2/ade2::URA3* |
| Strain, strain background (*Candida albicans*) | *frp1⁻/⁻ FRP1 URA3* | This work | KC1024 | KC870 *ADE2/ade2::FRP1 URA3* |
| Strain, strain background (*Candida albicans*) | *frp2⁻/⁻* | This work | KC1053 | KC590 *frp2Δ/frp2Δ* |
| Strain, strain background (*Candida albicans*) | *frp1⁻/⁻ frp2⁻/⁻* | This work | KC1061 | KC590 *frp1Δ/frp1D frp2Δ/frp2Δ* |
| Strain, strain background (*Candida albicans*) | *frp1⁻/⁻ FRP1p-FRP2* | This work | KC1064 | KC870 *ADE2/ade2::FRP1p-FRP2 URA3* |
| Strain, strain background (*Candida albicans*) | *SSB1p-FRP1* | This work | KC1080 | KC870 *ADE2/ade2::SSB1p-FRP1 URA3* |
| Strain, strain background (*Candida albicans*) | *frp1⁻/⁻ URA3* | This work | KC1146 | KC923 *ADE2/ade2::URA3* |
| Strain, strain background (*Candida albicans*) | *SSB1p-FRP2* | This work | KC1244 | KC912 *ADE2/ade2::SSB1p-FRP2 URA3* |
| Strain, strain background (*Candida albicans*) | *FRP2-GFP* | This work | KC1245 | KC912 *frp2Δ/FRP2-GFP URA3* |
| Strain, strain background (*Candida albicans*) | *frp2⁻/⁻ URA3* | This work | KC1246 | KC912 *ADE2/ade2::URA3* |
| Strain, strain background (*Candida albicans*) | *frp2⁻/⁻ FRP2 URA3* | This work | KC1379 | KC912 *ADE2/ade2::FRP2 URA3* |

*Continued on next page*

*Continued*

| Reagent type (species) or resource | Designation | Source or reference | Identifiers | Additional information |
|---|---|---|---|---|
| Strain, strain background (*Candida albicans*) | *FRP2-GFP* | This work | KC1405 | KC2 *FRP2/FRP2-GFP URA3* |
| Strain, strain background (*Candida albicans*) | *frp1⁻/⁻ frp2⁻/⁻* | This work | KC1410 | KC2 *frp1Δ/frp1D frp2Δ/frp2Δ* |
| Strain, strain background (*Candida albicans*) | *frp2⁻/⁻ FRP2 URA3* | This work | KC1411 | KC913 *ADE2/ade2::FRP2 URA3* |
| Strain, strain background (*Candida albicans*) | *frp2⁻/⁻ URA3* | This work | KC1412 | KC923 *frp2Δ/frp2Δ::hisG-URA3-hisG* |
| Strain, strain background (*Candida albicans*) | *frp2⁻/⁻ URA3* | This work | KC1414 | KC913 *ADE2/ade2::URA3* |
| Strain, strain background (*Candida albicans*) | *frp2⁻/⁻ FRP1p-FRP2* | This work | KC1447 | KC912 *ADE2/ade2:: FRP1p-FRP2 URA3* |
| Strain, strain background (*Sandida cerevisiae*) | *hem1* | This work | KY1498 | *ura3-1 can1-100 GAL+leu2-3,112 trp1-1 ade2-1 his3-11,15 hem1Δ::KanMX* |
| Antibody | Anti-GFP (Rabbit polyclonal) | Abcam | Cat # ab290 | Use at 1:5000 |
| Antibody | Anti-rabbit IgG, HRP-conjugated (Goat polyclonal) | Sigma | Cat # A9169 | Use at 1:5000 |
| Recombinant DNA reagent | *FRP1* blaster plasmids | This work | KB2392 KC2393 | Digest SacI-KpnI for transformation |
| Recombinant DNA reagent | *FRP2* blaster plasmids | This work | KB2395 KB2396 | Digest SacI-KpnI for transformation |
| Recombinant DNA reagent | *FRP1* reintegrant | This work | KB2546 | Contains the *FRP1* region from –395 to +1877 |
| Recombinant DNA reagent | *FRP2* reintegrant | This work | KB2576 | Contains the *FRP2* region from –895 to +1982 |
| Recombinant DNA reagent | *FRP1*-GFP | This work | KB2431 | Contains *FRP1* (+3 to+1662) fused to eGFP |
| Recombinant DNA reagent | *FRP2*-GFP | This work | KB2695 | Contains *FRP2* (–969 to +1776) fused to eGFP |
| Recombinant DNA reagent | *FRP1p-FRP2* | This work | KB2575 | Contains the *FRP1* promoter (–395 to –1) fused to the *FRP2* (+1 to +1878) |
| Recombinant DNA reagent | *SSB1p-FRP1* | This work | KB2450 | Contains *FRP1* (–2 to +1826) under *SSB1* promoter (–400 to –1) |
| Recombinant DNA reagent | *SSB1p-FRP2* | This work | KB2696 | Contains *FRP2* (–3 to +1933) under *SSB1* promoter (–400 to –1) |
| Recombinant DNA reagent | *PGA7 FRP1* | This work | KB2566 | *H2Ap-PGA7, H2Bp-FRP1 codon corrected* |
| Recombinant DNA reagent | *FRP1* | This work | KB2569 | *H2Bp-FRP1 codon corrected* |
| Recombinant DNA reagent | *PGA7* | This work | KB2789 | *H2Ap-PGA7* |

*Continued*

| Reagent type (species) or resource | Designation | Source or reference | Identifiers | Additional information |
|---|---|---|---|---|
| Sequence-based reagent | *FRP1* codon-corrected | This work | B35984-1/M131416 | *FRP1* codon-corrected in pUC57 |
| Sequence-based reagent | SacI-FRP1 5′ (−795) | This work | PCR primer 1503 | GCGAGCTCCCAGCAGCACTTCCTG |
| Sequence-based reagent | FRP1 5′ (−1) – SpeI | This work | PCR primer 1504 | ggactaGTTGAAAGTTAAACTTGGTTA |
| Sequence-based reagent | HindIII-FRP1 3′ (+1662) | This work | PCR primer 1505 | GGGAAGCTTAGGGTATATAGGATAAAT |
| Sequence-based reagent | FRP1 3′ (+2258) - KpnI | This work | PCR primer 1506 | gcggtACCCAAATGCATGGGTAAAC |
| Sequence-based reagent | FRP1 test (−810) | This work | PCR primer 1507 | CACTTGCACTACCAGTTTCG |
| Sequence-based reagent | SacI-FRP2 5′ (−730) | This work | PCR primer 1508 | GGGAGCTCGGAAAATAAGTTGTTCTTTG |
| Sequence-based reagent | FRP2 5′ (+3) – SpeI | This work | PCR primer 1509 | CGACTAGTCCATGGCTGATAAGTTG |
| Sequence-based reagent | HindIII-FRP2 3′ (+1753) | This work | PCR primer 1510 | GGGAAGCTTCTATAACGAGTCGTACGA |
| Sequence-based reagent | FRP2 3′ (+2394) – KpnI | This work | PCR primer 1511 | CCGGTACCTGATCCTTGGATGCCA |
| Sequence-based reagent | FRP2 test (−750) | This work | PCR primer 1512 | GTAACAAACCCGAGAACACC |
| Sequence-based reagent | RI-FRP1 | This work | PCR primer 1513 | CCGAATTCAACCATGGCTATTCCAT |
| Sequence-based reagent | FRP1(+1740)-XhoI | This work | PCR primer 1514 | cgctcgaGGTGTGTCCTTACGTACAG |
| Sequence-based reagent | RI-FRP2 | This work | PCR primer 1515 | gcGAATTCCATGGACGAAGAACTTCAG |
| Sequence-based reagent | FRP2 (+1872)-XhoI | This work | PCR primer 1516 | ggctcGAGAGTGCTGTGAGGTTATG |
| Sequence-based reagent | BamHI-FRP1(+3) | This work | PCR primer 1522 | gcggatccgctattccatttgatcaacag |
| Sequence-based reagent | FRP1(+1662)-XhoI | This work | PCR primer 1523 | cgctcgagaaacgactctgtataacaatac |
| Sequence-based reagent | SpeI-FRP1 5′ (−395) | This work | PCR primer 1541 | ccACTAGTCGTAATCAGCAGCAGATAC |
| Sequence-based reagent | FRP1 3′ (+1877) - KpnI | This work | PCR primer 1542 | ccGGTACCGCACAAGCGGGTACT |
| Sequence-based reagent | FRP1 (+1550)-XhoI | This work | PCR primer 1543 | GGCTCGAGTTTGTGAATGATGGCGA |
| Sequence-based reagent | PGA7 (+1087)-HindIII | This work | PCR primer 1544 | GCAAGCTTGGCATACTCAATTTGATG |
| Sequence-based reagent | 5′-F1HEM1 | This work | PCR primer | CCCTCAATAATCATAACAGTACTTAGGTTTTTTTTTCAGTCGGATCCCCGGGTTAATTAA |
| Sequence-based reagent | 3′-R1HEM1 | This work | PCR primer | CCTTGTACCTCTATCTCAGCCCATGCATATATTGGTTGTTGAATTCGAGCTCGTTTAAAC |
| Sequence-based reagent | Promoter 5′ (HTA2) | This work | PCR primer | TATATATTAAATTTGCTCTTGTTC |

*Continued on next page*

*Continued*

| Reagent type (species) or resource | Designation | Source or reference | Identifiers | Additional information |
|---|---|---|---|---|
| Sequence-based reagent | Promoter 3′ (HTB2) | This work | PCR primer | TAGTTGTAGAGTAAGTTGTTG |
| Sequence-based reagent | SacI-HTB2p –684 | This work | PCR primer | gcgaGCTCTTGTTCTGTACTTTCC |
| Sequence-based reagent | PGA7 5′ | This work | PCR primer | GAACAAGAGCAAATTTAATATATAATGCATTTCATATTCTACTTGA |
| Sequence-based reagent | Pga7-SacI (+697) | This work | PCR primer | Pga7-SacI (+697) |
| Sequence-based reagent | FRP1 5′ | This work | PCR primer | CAACAACTTACTCTACAACTAATGGCTATTCCATTTGATCAA |
| Sequence-based reagent | FRP1-SalI | This work | PCR primer | CcccgtcgacggtatcgA |
| Chemical compound, drug | Hemin | Frontier Scientific | H651-9 | Hemin chloride |
| Chemical compound, drug | Bovine hemoglobin | Sigma-Aldrich | H2500 | |
| Chemical compound, drug | Ferrozine | Sigma-Aldrich | P9762 | 3-(2-Pyridyl)–5,6-diphenyl-1,2,4-triazine-4′,4″-disulfonic acid sodium salt |
| Chemical compound, drug | BPS | Sigma-Aldrich | B1375 | Bathophenanthroline sulfonate |
| Chemical compound, drug | ALA | Merck | 08339 | δ-Aminolevulinic acid |
| Chemical compound, drug | GaPPIX | Frontier Scientific | P40167 | $Ga^{3+}$-protoporphyrin IX chloride |
| Chemical compound, drug | CoPPIX | Frontier Scientific | Co654-9 | $Co^{3+}$-protoporphyrin IX chloride |
| Chemical compound, drug | MnPPIX | Frontier Scientific | MnP562-9 | $Mn^{3+}$-protoporphyrin IX chloride |
| Chemical compound, drug | ZnPPIX | Frontier Scientific | Zn625-9 | $Zn^{2+}$-protoporphyrin IX |
| Chemical compound, drug | ZnMP | Chem-Cruz | Sc-396862 | $Zn^{2+}$-mesoporphyrin |
| Other | Alignment summary of 40 saccharomycetales species for phylogenetic profiling | This work | | github.com/BKU-Technion/FRP |

## Media and chemicals

Cells were grown in YPD medium (1% yeast extract, 2% bacto-peptone, 2% glucose, tryptophan 150 mg/l) or in Synthetic Complete (SC) medium lacking specific amino acids, as indicated. SC medium contains, per liter, Yeast Nitrogen Base (USBiological) 1.7 g, $(NH_4)_2SO_4$ 5 g, the 20 amino acids, adenine and uridine, 0.1 g each except leucine, 0.2 g, glucose 20 g, and 0.2 mM inositol. YPD pH 7.5 or pH 8.5 were made by adding 0.1 M Tris-Cl from a 1 M stock of appropriate pH. Media were supplemented with the iron chelators ferrozine or BPS (Sigma) at 1 mM, hemin as indicated from a 2 mM stock in 50 mM NaOH, or bovine hemoglobin from a 0.5 mM stock in phosphate-buffered saline (Dulbecco's PBS; Biological Industries, Israel). ALA (Merck) was added to 50 μg/ml in the medium to maintain the *hem1* cells. Hemin, $Ga^{3+}$-protoporphyrin IX, $Co^{3+}$-protoporphyrin IX, $Mn^{3+}$-protoporphyrin IX, $Zn^{2+}$-protoporphyrin IX, and ZnMP were obtained from Frontier Scientific, and bovine hemoglobin from Sigma (H2500). ZnMP was added from a 6 mM stock made fresh in 10% ethanolamine, and $Ga^{3+}$-protoporphyrin IX, $Co^{3+}$-protoporphyrin IX, $Mn^{3+}$-protoporphyrin IX, $Zn^{2+}$-protoporphyrin IX were added from a 50 mM stock made fresh in DMSO.

**Table 1.** List of *Candida albicans* strains.

| Name | Genotype | Origin |
|------|----------|--------|
| KC2=CAF3-1 | *ura3Δ::imm434/ura3Δ::imm434* | *Fonzi and Irwin, 1993* |
| KC68 | KC2 *ccc2Δ::hisG/ccc2Δ::hisG* | *Weissman et al., 2002* |
| KC139 | KC68 *rbt5Δ/rbt5Δ* | *Weissman and Kornitzer, 2004* |
| KC485 | KC68 *pga7Δ/pga7Δ* | *Kuznets et al., 2014* |
| KC590=CAI4 | *ura3Δ::imm434/ura3Δ::imm434* | *Fonzi and Irwin, 1993* |
| KC646 | KC590 *pga7Δ/pga7Δ ADE2/ade2::URA3* | *Kuznets et al., 2014* |
| KC647 | KC590 *pga7Δ/pga7Δ ADE2/ade2::PGA7 URA3* | *Kuznets et al., 2014* |
| KC811 | KC68 *ADE2/ade2::URA3* | This work |
| KC859 | KC2 *FRP1/frp1Δ* | This work |
| KC870 | KC2 *frp1Δ/frp1Δ* | This work |
| KC901 | KC2 *FRP2/frp2Δ* | This work |
| KC912 | KC2 *frp2Δ/frp2Δ* | This work |
| KC913 | KC68 *frp2Δ/frp2Δ* | This work |
| KC914 | KC2 *FRP1/FRP1-GFP URA3* | This work |
| KC916 | KC2 *frp1Δ/FRP1-GFP URA3* | This work |
| KC923 | KC68 *frp1Δ/frp1Δ* | This work |
| KC966 | KC590 *frp1Δ/frp1Δ* | This work |
| KC1023 | KC870 *ADE2/ade2::URA3* | This work |
| KC1024 | KC870 *ADE2/ade2::FRP1 URA3* | This work |
| KC1053 | KC590 *frp2Δ/frp2Δ* | This work |
| KC1061 | KC590 *frp1Δ/frp1Δ frp2Δ/frp2Δ* | This work |
| KC1064 | KC870 *ADE2/ade2::FRP1p-FRP2 URA3* | This work |
| KC1080 | KC870 *ADE2/ade2::SSB1p-FRP1 URA3* | This work |
| KC1146 | KC923 *ADE2/ade2::URA3* | This work |
| KC1244 | KC912 *ADE2/ade2::SSB1p-FRP2 URA3* | This work |
| KC1245 | KC912 *frp2Δ/FRP2-GFP URA3* | This work |
| KC1246 | KC912 *ADE2/ade2::URA3* | This work |
| KC1379 | KC912 *ADE2/ade2::FRP2 URA3* | This work |
| KC1405 | KC2 *FRP2/FRP2-GFP URA3* | This work |
| KC1410 | KC2 *frp1Δ/frp1Δ frp2Δ/frp2Δ* | This work |
| KC1411 | KC913 *ADE2/ade2::FRP2 URA3* | This work |
| KC1412 | KC923 *frp2Δ/frp2Δ::hisG-URA3-hisG* | This work |
| KC1414 | KC913 *ADE2/ade2::URA3* | This work |
| KC1447 | KC912 *ADE2/ade2:: FRP1p-FRP2 URA3* | This work |

## Strains and plasmids

*Candida albicans* strains used are listed in *Table 1*. The *FRP1* and *FRP2* 'blaster' deletion plasmids, KB2392, -3 and KB2395, -6 were built by introducing 500 nt adjacent to the 5' and 3' ends of each gene into plasmids KB985 and KB986 (*Atir-Lande et al., 2005*). They were used to build strains KC859, KC870, KC901, KC912, KC913, KC923, KC966, KC1053, KC1061, KC1410, KC1412. KB2546

contains the *FRP1* region (from –395 to +1877) cloned SpeI-KpnI into BES116 (*Feng et al., 1999*). It was used to make the *FRP1* reintegrant strain KC1024. KB2576 contains the *FRP2* region (from –895 to +1982) cloned SpeI-ApaI into BES116. It was used to make the *FRP2* reintegrant strains KC1379, 1411. Vector BES116 was used to make strains KC811, 1023, 1146, 1246, 1414. KB2431 contains *FRP1* (+3 to +1662, BamHI-XhoI) and KB2695 contains *FRP2* (–969 to +1776, SpeI-XhoI) fused to the eGFP sequence of KB2430 (*Bar-Yosef et al., 2018*). They were used to construct KC914+KC916 and KC1245+KC1405, respectively. KC2575 contains the *FRP1* promoter (–395 to –1) cloned in BES116 and then fused to the *FRP2* open reading frame and 3' region (+1 to +1878). It was used to make KC1064, KC1447. KB2450 and KB2696, containing *FRP1* (–2 to +1826) and *FRP2* (–3 to +1933), respectively, cloned SpeI-KpnI in KB2448, were used to make KC1080 and KC1244, respectively. KB2448 is BES116 with the *SSB1* promoter region (–400 to –1) cloned NotI-SpeI. *S. cerevisiae* strain KY1498 is *ura3-1 can1-100 GAL+leu2-3,112 trp1-1 ade2-1 his3-11,15 hem1Δ::KanMX*. It was constructed by transforming strain W303-1A (*Wallis et al., 1989*) with a KanMX marker amplified from pFA6-kanMX6 (*Longtine et al., 1998*) using primers with 40 bp homology at each end to *HEM1* adjacent sequences. To express *FRP1* and *PGA7* in *S. cerevisiae*, we first ordered a synthetic version of the *FRP1* coding sequence (Hy-labs, Rehovot, Israel) with all ambiguous CTG codons changed to other leucine codons. We then amplified this gene, *PGA7*, and the promoter region located between the 5' ends of the adjacent *S. cerevisiae* histone genes *HTA2* and *HTB2*. Fusion PCR was used to fuse the three fragments, which were then cloned SacI-SalI in the vector p413GAL1 (*Mumberg et al., 1994*) (removing the *GAL1* promoter) to generate KB2566. The *HTA2-HTB2* promoter fragment was also fused to *FRP1* alone and cloned likewise to generate KB2569. To generate KB2789, *FRP1* was removed from KB2566 by digestion with EcoRV-NcoI, filling in and re-ligation.

## Growth assays

For hemoglobin utilization, overnight cultures grown in YPD were diluted in the morning into a series of twofold dilutions of hemoglobin in YPD+ferrozine or BPS. Cells were inoculated in flat-bottomed 96-well plates at $OD_{600}$=0.00001, 150 µl per well. Plates were incubated at 30°C on an orbital shaker at 60 rpm and growth was measured by optical density ($OD_{600}$) after 2 and 3 days with an ELISA reader. Cells were resuspended with a multi-pipettor before each reading. Each culture was done in triplicate. For MPP sensitivity, cells were inoculated in flat-bottomed 96-well plates in YPD+1 mM ferrozine at $OD_{600}$=0.0001, 150 µl per well. Plates were incubated at 30°C on an orbital shaker at 60 rpm and growth was measured by optical density ($OD_{600}$) after 2 days with an ELISA reader. All experiments shown were performed more than once with similar results. The heme-iron utilization phenotypes were tested in different background strains ($CCC2^{+/+}$, $ccc2^{-/-}$) with similar results.

## Heme sensor-binding assays

The protocol was as described (*Weissman et al., 2021*). Briefly, each strain to be tested was transformed with the sensor plasmids KB2636 (WT HS1), KB2669 (M7A), as well as a vector plasmid. Overnight cultures were diluted to $OD_{600} = 0.2$ and grown for 4 hr in the indicated hemin concentrations. For fluorescence measurements, cells were pelleted, washed once with PBS, resuspended to $OD_{600} = 5$, and 0.2 ml were placed in duplicate in a black 96-well flat-bottom plates (Nunc Fluorotrac). Fluorescence was measured with a Tecan infinite 200 Pro reader, with eGFP: ex. 480 nm (9 nm bandwidth), em. 520 nm (20 nm bandwidth), and mKATE2 ex. 588 nm (9 nm bandwidth), em. 620 nm (20 nm bandwidth). Each strain was represented by three independent cultures, and each culture was measured twice (technical duplicate), the vector-only culture reading was substracted from all other readings, and the ratio of eGFP to mKATE2 was calculated. The experiment shown was preceded by four experiments with different strain combinations and media, all showing reduced heme intake in the $frp1^{-/-}$ and $frp2^{-/-}$ mutants compared to the wild-type. Statistical analysis: To test the effect of genotype and hemin concentration on the ratio, we used two-way ANOVA followed by Tukey's honestly significant difference test, which controls for multiple comparisons. For each clone we report the results of pairwise comparisons between each hemin concentration and no hemin. The analysis was performed using R.

## Microscopy

Cells were imaged with a Zeiss Axioskop Imager epifluorescence microscope equipped with DIC optics, using a ×100 objective. For visualizing Frp1-GFP and Frp2-GFP, the GFP filter set was used with an exposure time of 6 s. Cells were inoculated a day before in YPD medium to reach late-log phase ($OD_{600}$ = 5–8) in the morning before induction. For quantitation of subcellular localization, at least 100 cells were scored for each datapoint and each cell was given a score between 0 and 3 for each location. The scoring was done blind, on pictures taken by a different operator and given an arbitrary coding. The experiment shown was performed after a preliminary experiment with fewer timepoints that yielded similar results.

*ZnMP uptake*: Cells were grown overnight under iron starvation (YPD+1 mM ferrozine). The next morning cells were diluted 1:50 in 5 ml of the same medium and grown for 3–4 hr. One μM ZnMP was added. Samples were taken after 10 min and spun down for 10 s at maximum speed. The supernatant was removed, and the pellet was once washed with 1 ml of ice-cold 2% BSA in PBS, then resuspended in 30 μl ice- cold 2% BSA in PBS. For visualizing ZnMP by epifluorescence microscopy, the rhodamine filter was used with an exposure time of 10 s.

*CMAC vacuolar stain*: CMAC was added to a logarithmically growing culture to 5 μM final from a 10 mM stock solution in DMSO and the cells were incubated a further 30 min at 30°C prior to microscopy visualization. For visualizing CMAC by epifluorescence microscopy, the DAPI filter was used with an exposure time of 700 ms.

*Hoechst 33342 nuclear stain*: Hoechst 33342 was added to a logarithmically growing culture to 10 μg/ml from a 1 mg/ml stock in water and the cells were incubated a further 30 min at 30°C prior to microscopy visualization. For visualizing Hoechst 33342 by epifluorescence microscopy, the DAPI filter was used with an exposure time of 1 s.

## Protein analysis

Protein extracts were obtained by resuspending three $OD_{600}$ of cells in 120 μM of protein loading buffer, mixing the suspension with glass beads and shaking the suspension in a bead-beater (Next Advance Bullet Blender 24) for 5 min at 4°C. The supernatants were heated to 37°C for 5 min and loaded on Bio-Rad 4–20% pre-cast mini-gels. After Western transfer, the membranes were reacted with a rabbit anti-GFP polyclonal antibody (Abcam ab290) and with an anti-rabbit horseradish peroxidase-conjugated secondary antibody (Sigma A9169). Luminescence signals were visualized with a Vilber Fusion FX Spectra imager.

## Phylogenetic profiling

*Building the profiles*: we performed a BLAST search (tblastn) with all *C. albicans* proteins as query and 40 Saccharomycotina genomes (reflecting 40 species) downloaded from the Joint Genome Institute site (https://mycocosm.jgi.doe.gov/mycocosm/home) and the Candida Genome Database site (http://www.candidagenome.org/) in May 2015. An alignment summary of each *C. albicans* protein with each of the 40 other species can be downloaded from https://github.com/BKU-Technion/FRP; *BKU-Technion, 2022*. For each protein we recorded the best score in each genome. We first looked at the profiles of Pga7/Rbt5/Csa2 and found that they are detectable in nine species out of this set of 40, including the more distant *Blastobotrys* (*Arxula*) *adeninivorans* and *Trichomonascus petasosporus* but excluding the more closely related *Candida guilliermondii*. We next looked at patterns of conservation of homologous proteins that mirror those of the CFEM proteins, that is, that are present in all nine species containing CFEM proteins and absent from all other genomes. Only Frp1/2 fulfilled this condition, that is, presence in the nine species containing Pga7/Rbt5/Csa2 homologs and absence in the other species, including *C. guilliermondii*.

## Materials availability statement

All strains and all plasmids are available upon request with no restrictions.

## Acknowledgements

We thank J Becker (U of Tennessee) for strain KC590, and Sara Selig for critical reading of the manuscript. This study was supported by ISF grant 587/19 to DK.

## Additional information

### Funding

| Funder | Grant reference number | Author |
| --- | --- | --- |
| Israel Science Foundation | 587/19 | Daniel Kornitzer |

The funders had no role in study design, data collection and interpretation, or the decision to submit the work for publication.

### Author contributions

Udita Roy, Validation, Investigation, Visualization, Methodology, Writing - original draft; Shir Yaish, Investigation, Visualization, Methodology; Ziva Weissman, Supervision, Investigation, Visualization, Methodology; Mariel Pinsky, Investigation, Visualization; Sunanda Dey, Investigation; Guy Horev, Software, Formal analysis; Daniel Kornitzer, Conceptualization, Formal analysis, Supervision, Funding acquisition, Visualization, Writing - original draft, Project administration, Writing - review and editing

### Author ORCIDs

Daniel Kornitzer http://orcid.org/0000-0002-5062-2735

### Decision letter and Author response

Decision letter https://doi.org/10.7554/eLife.80604.sa1
Author response https://doi.org/10.7554/eLife.80604.sa2

## Additional files

### Supplementary files

• Supplementary file 1. List of Alphafold 2-predicted *Candida albicans* protein structures most similar to the Csa2 CFEM hemophore domain structure.

• MDAR checklist

### Data availability

All data generated or analyzed during this study are included in the manuscript and supporting files. A sequence alignment summary file used for phylogenetic profiling analysis is available on Github at https://github.com/BKU-Technion/FRP, (copy archived at swh:1:rev:5de8de2829b273b0ec4b5741b15061fd27c27231).

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
