## [Editor Report]

This work focuses on the important problem of how a human pathogenic fungus obtains iron during infection. This study uses biochemical and genetic methods to identify the missing link between heme receptors and heme utilization by cells. Specifically, they provide convincing evidence that the ferric reductase-like proteins Frp1 and Frp2 have major roles in iron acquisition from heme, thus identifying a new function for these proteins. Understanding how pathogenic fungi obtain iron and identifying differences in fungal and host iron metabolism can provide valuable leads for drug discovery.

---

## [Decision Letter]

**Decision letter after peer review:**

Thank you for submitting your article "Ferric reductase-related proteins mediate fungal heme acquisition" for consideration by *eLife*. Your article has been reviewed by 2 peer reviewers, and the evaluation has been overseen by a Reviewing Editor and Arturo Casadevall as the Senior Editor. The following individual involved in the review of your submission has agreed to reveal their identity: Valeria C Culotta (Reviewer #1).

Essential revisions:

1) Please address the issues raised by Reviewers 1 and 2.

2) Reviewer 2 raised the question of the direct transfer of heme from Pga7 to Frp1. We all agree that this would be very important issue if it could be addressed. However, we recognize that it would require a considerable amount of new work and we are not requiring it. That said, anything you could do to address this point would strengthen the manuscript.

*Reviewer #1 (Recommendations for the authors):*

Suggestions for improvement:

1) There appears to be a discrepancy in results with Figure 1 and Figure 2. In Figure 1, Frp1 is clearly dominant for utilization of heme as an Fe source at both pH. In Figure 2, Frp2 is dominant for heme uptake by the sensor, although studies were only conducted at neutral pH. These apparent discrepancies are not a simple matter of pH because, in Fig, 1, Frp1 is still dominant at neutral pH. Could there be two functions, ie., heme utilization where Frp1 is dominant and secondly, heme uptake where Frp2 is dominant? This should be discussed.

2) Statistics are needed for the graphs of Figure 4B and Figure 5B. In Figure 4B, 100 cells were counted. Was that from one experimental trial, explaining why no error bars?

3) Figure 1C – the data with pga7 is impossible to see. Is there a way to change the icon size so that pga7 is visible together with frp1?

4) In the sentence: The reduced increase in sensor occupancy in the frp2-/- mutant exposed to external hemin compared to the frp1-/- mutant is consistent with the growth data obtained at pH 8.5 (Figure 2), Shouldn't this be Figure 1 since Figure 1 is the growth data?

5) The vacuolar signal of the GFP proteins looks luminal. If the authors think this is degraded Frp1 and Frp2, they should mention this.

*Reviewer #2 (Recommendations for the authors):*

The manuscript is very clearly written and the experiments are presented in a very deliberate sequence that makes for a very clear set of studies. The only question I would like to see addressed is whether there is a direct transfer of heme from Pga7 to Frp1. The authors suggest that reduction of the bound heme could affect its release and that implies that the reductase has direct interaction with the heme bound on Pga7. Perhaps proximity labeling or proximity ligation approaches could be used to answer this question. An argument could be made that these studies are complete without this question being mechanistically addressed.

---

## [Author Response]

Essential revisions:Reviewer #1 (Recommendations for the authors):Suggestions for improvement:1) There appears to be a discrepancy in results with Figure 1 and Figure 2. In Figure 1, Frp1 is clearly dominant for utilization of heme as an Fe source at both pH. In Figure 2, Frp2 is dominant for heme uptake by the sensor, although studies were only conducted at neutral pH. These apparent discrepancies are not a simple matter of pH because, in Fig, 1, Frp1 is still dominant at neutral pH. Could there be two functions, ie., heme utilization where Frp1 is dominant and secondly, heme uptake where Frp2 is dominant? This should be discussed.

Thank you for pointing this out. Indeed, Frp2 appears to be more dominant in the sensor assay (heme uptake) but is less dominant in the growth assays (heme utilization). We have rewritten the description of the experiment (lines 208-210 in the revision), and added a possible explanation for the discrepancy in the Discussion (lines 440-445).

2) Statistics are needed for the graphs of Figure 4B and Figure 5B. In Figure 4B, 100 cells were counted. Was that from one experimental trial, explaining why no error bars?

Regarding Figure 4B, preliminary experiments had consistently show cell surface relocalization of the Frp proteins in the presence of hemin or hemoglobin. In the experiment shown, this relocalization was quantitated, and a parallel experiment, identical to the one we showed except that the cells were exposed to hemin rather than to hemoglobin, showed very similar kinetics of cell surface relocalization of the signal and concomitant decrease in ER signal. However, in the course of testing the nature of the vacuolar Frp1/2-GFP signals as prompted by comment #5, we found that, while the shift of the GFP signal to the cell surface was robustly reproducible, the decrease in ER signal was not. This might be due to a different induction regime: rather than using cultures that had already been grown overnight under iron limitation, as we had in previous experiments, in the new experiments we shifted log-phase cells growing in YPD to iron-limited medium, in order to monitor induction from a low expression baseline. In any case, increase in cell surface signal is not always accompanied by decrease in the ER signal. We have thus modified the text accordingly. New panels B and C of Figure 4 show induction of Frp1 and Frp2-GFP without and with hemin, and kinetics of appearance of the signal on the cell surface, respectively. (The new experiments uniformly use hemin rather than hemoglobin, since previous experiments indicated that both compounds induce similar changes in Frp1 and Frp2 subcellular distribution).

Regarding statistics, for the graphs in Figure 4C and Figure 5B, we have now added asterisks indicating statistically significant differences from time 0’ and from vector control, respectively.

3) Figure 1C – the data with pga7 is impossible to see. Is there a way to change the icon size so that pga7 is visible together with frp1?

Thanks – we have made the *pga7* icon bigger so that it is now visible in all graphs.

4) In the sentence: The reduced increase in sensor occupancy in the frp2-/- mutant exposed to external hemin compared to the frp1-/- mutant is consistent with the growth data obtained at pH 8.5 (Figure 2), Shouldn't this be Figure 1 since Figure 1 is the growth data?

Indeed. Fixed, thanks.

5) The vacuolar signal of the GFP proteins looks luminal. If the authors think this is degraded Frp1 and Frp2, they should mention this.

Thanks for bringing this up. We had indeed assumed that the luminal signal represented remnants of the Frp1-GFP and Frp2-GFP proteins, the GFP moiety being notoriously slower to degrade in the vacuole. However, this comment prompted us to test this notion directly by Western blotting. As can be seen in new Figure 4 —figure supplement 2, only the full-length fusion proteins are detectable by Western. This is true whether in the presence of hemin, where more of the signal is on the cell surface, or in its absence, where the vacuolar signal is more prominent, or whether upon shifting the induced cells to iron satiety, which causes degradation of the fusion proteins. This suggests that there is no accumulation of GFP degradation fragments in the cell, and that all the visible GFP signal represents full-length, potentially active protein.

Reviewer #2 (Recommendations for the authors):The manuscript is very clearly written and the experiments are presented in a very deliberate sequence that makes for a very clear set of studies. The only question I would like to see addressed is whether there is a direct transfer of heme from Pga7 to Frp1. The authors suggest that reduction of the bound heme could affect its release and that implies that the reductase has direct interaction with the heme bound on Pga7. Perhaps proximity labeling or proximity ligation approaches could be used to answer this question. An argument could be made that these studies are complete without this question being mechanistically addressed.

Thank you. We couldn’t agree more regarding the importance of the questions raised by the reviewer. These questions form a main section of our current research program. Regarding interaction between Frp1 and Pga7 – which could be transient –, we have already attempted co-i.p. of the proteins with or without crosslinking – so far without success. Proximity labeling is something we hadn’t considered, we will now. Regarding direct heme transfer from the CFEM hemophores to Frp1/2, demonstrating such a transfer is technically much more challenging than showing heme transfer between the CFEM hemophores, because Frp1/2 are not soluble proteins, and because they already contain two catalytic hemes in their transmembrane domain. Therefore, we are currently focusing on identifying the heme receptor site on Frp1, and, with collaborators, on demonstrating the predicted heme reductase activity of Frp1/2.